# Large-scale replication study reveals a limit on probabilistic prediction in language comprehension

Mante S Nieuwland[1,2]*, Stephen Politzer-Ahles[3,4], Evelien Heyselaar[5], Katrien Segaert[5], Emily Darley[6], Nina Kazanina[6], Sarah Von Grebmer Zu Wolfsthurn[6], Federica Bartolozzi[2], Vita Kogan[2], Aine Ito[2,4], Diane Mézière[2], Dale J Barr[7], Guillaume A Rousselet[7], Heather J Ferguson[8], Simon Busch-Moreno[9], Xiao Fu[9], Jyrki Tuomainen[9], Eugenia Kulakova[10], E Matthew Husband[4], David I Donaldson[11], Zdenko Kohút[12], Shirley-Ann Rueschemeyer[12], Falk Huettig[1]

[1]Max Planck Institute for Psycholinguistics, Nijmegen, Netherlands; [2]School of Philosophy, Psychology and Language Sciences, University of Edinburgh, Edinburgh, United Kingdom; [3]Department of Chinese and Bilingual Studies, The Hong Kong Polytechnic University, Kowloon, Hong Kong; [4]Faculty of Linguistics, Philology and Phonetics, University of Oxford, Oxford, United Kingdom; [5]School of Psychology, University of Birmingham, Birmingham, United Kingdom; [6]School of Experimental Psychology, University of Bristol, Bristol, United Kingdom; [7]Institute of Neuroscience and Psychology, University of Glasgow, Glasgow, United Kingdom; [8]School of Psychology, University of Kent, Canterbury, United Kingdom; [9]Division of Psychology and Language Sciences, University College London, London, United Kingdom; [10]Institute of Cognitive Neuroscience, University College London, London, United Kingdom; [11]Psychology, Faculty of Natural Sciences, University of Stirling, Stirling, United Kingdom; [12]Department of Psychology, University of York, York, United Kingdom

*For correspondence:
mante.nieuwland@mpi.nl

Competing interests: The authors declare that no competing interests exist.

**Abstract** Do people routinely pre-activate the meaning and even the phonological form of upcoming words? The most acclaimed evidence for phonological prediction comes from a 2005 *Nature Neuroscience* publication by DeLong, Urbach and Kutas, who observed a graded modulation of electrical brain potentials (N400) to nouns and preceding articles by the probability that people use a word to continue the sentence fragment ('cloze'). In our direct replication study spanning 9 laboratories (*N*=334), pre-registered replication-analyses and exploratory Bayes factor analyses successfully replicated the noun-results but, crucially, not the article-results. Pre-registered single-trial analyses also yielded a statistically significant effect for the nouns but not the articles. Exploratory Bayesian single-trial analyses showed that the article-effect may be non-zero but is likely far smaller than originally reported and too small to observe without very large sample sizes. Our results do not support the view that readers routinely pre-activate the phonological form of predictable words.
DOI: https://doi.org/10.7554/eLife.33468.001

## Introduction

In the last decades, the idea that people routinely and implicitly predict upcoming words during language comprehension turned from a highly controversial hypothesis to a widely accepted

assumption. Initial objections to prediction in language were based on a lack of empirical support (e.g. *Zwitserlood, 1989*), incompatibility with traditional bottom-up models and contemporary interactive models of language comprehension (e.g. *Kintsch, 1988*; *Marslen-Wilson and Tyler, 1980*), and the purported futility of prediction in a generative system where sentences can continue in infinitely many different ways (*Jackendoff, 2002*). Current theories of language comprehension, however, reject such objections and posit prediction as an integral and inevitable mechanism by which comprehension proceeds quickly and incrementally (e.g. *Altmann and Mirković, 2009*; *Dell and Chang, 2014*; *Pickering and Garrod, 2013*). Prediction, that is, context-based pre-activation of an upcoming linguistic input, is thought to occur at all levels of linguistic representation (semantic, morpho-syntactic and phonological/orthographic) and serves to facilitate the integration of newly available bottom-up information into the unfolding sentence- or discourse-representation. In this line of thought, language is yet another domain in which the brain acts as a prediction machine (*Clark, 2013*; *Van Berkum, 2010*; see also *Friston, 2005*, *2010*; *Summerfield and de Lange, 2014*), hard-wired to continuously match sensory inputs with top-down, grammatical or probabilistic expectations based on context and memory.

What promoted linguistic prediction from outlandish and deeply contentious to ubiquitous and somewhat anodyne? One of the key and most acclaimed pieces of empirical evidence for linguistic prediction to date comes from a landmark *Nature Neuroscience* publication by *DeLong et al. (2005)*, whose approach exploited a phonological rule of English whereby the indefinite article is realized as *a* before consonant-initial words and as *an* before vowel-initial words. In their experiment, participants read sentences of varying degree of contextual constraint that led to expectations for a particular consonant- or vowel-initial noun. This expectation was operationalized as a word's cloze probability (cloze), calculated in a separate, non-speeded sentence completion task as the percentage of continuations of a sentence fragment with that word (*Taylor, 1953*). For example, the sentence fragment "The day was breezy so the boy went outside to fly..." is continued with 'a' by 86% of participants, and "The day was breezy so the boy went outside to fly a..." is continued with 'kite' by 89% of participants. In the main experiment, word-by-word sentence presentation enabled DeLong and colleagues to examine electrical brain activity elicited by articles that were concordant with the highly expected but yet unseen noun ('a', followed by 'kite'), or by articles that were incompatible with the highly expected noun and heralded a less expected one ('an', followed by 'airplane'). Of note, an unexpected like 'a/an' does not rule out that the expected noun appears, just that it appears as the immediately following word (e.g., 'an old kite'), we return to this issue in the Discussion. The dependent measure was the amplitude of the N400 Event-related potential (ERP), a negative ERP deflection that peaks approximately 400 ms after word onset and is maximal at centroparietal electrodes (*Kutas and Hillyard, 1980*). The N400 is elicited by every word of an unfolding sentence, and its amplitude is smaller (less negative) with increasing ease of semantic processing (*Kutas and Hillyard, 1984*). In this article, we use 'N400 amplitude' as a shorthand for 'ERP amplitude in the time window associated with the N400'; this ERP amplitude is actually a sum of the N400 ERP component and other ERP components (reflecting other aspects of cognition) that overlap with it in time and space. DeLong et al. found that the N400 amplitude for a given word decreased as a function of increasing cloze probability, both for nouns and, critically, for articles. DeLong et al. presented the systematic and graded N400 modulation by article-cloze as strong evidence that participants activated the nouns and articles in advance of their appearance, and that the disconfirmation of this prediction by the less-expected articles resulted in processing difficulty (higher N400 amplitude at the article).

The results obtained with this elegant design warranted a much stronger conclusion than related results available at the time. Previous studies that employed a visual-world paradigm had revealed listeners' anticipatory eye-movements toward visual objects on the basis of probabilistic or grammatical considerations (e.g. *Altmann and Kamide, 1999*). However, predictions in such studies are scaffolded onto already-available visual context, and therefore do not measure purely pre-activation, but perhaps re-activation of word information previously activated by the visual object itself (*Huettig, 2015*). DeLong and colleagues examined brain responses to information associated with concepts that were not pre-specified and had to be retrieved from long-term memory 'on-the-fly'. Furthermore, DeLong and colleagues were the first to muster evidence for highly specific pre-activation of a word's phonological form, rather than merely its semantic (e.g. *Federmeier and Kutas, 1999*) or morpho-syntactic features (e.g. *Van Berkum et al., 2005*; *Wicha et al., 2004*). Crucially, as

their demonstration involved semantically identical articles (function words) rather than nouns or adjectives (content words) that are rich in meaning, the observed N400 modulation by article-cloze is unlikely to reflect difficulty interpreting the articles themselves. Most notably, DeLong and colleagues were the first to examine brain activity elicited by a range of more- or less-predictable articles, not simply most- versus least-expected. Based on the observed correlation, they argued that pre-activation is not all-or-none and limited to highly constraining contexts, but occurs in a graded, probabilistic fashion, with the strength of a word pre-activation proportional to its cloze probability. Moreover, they concluded that prediction is an integral part of real-time language processing and, most likely, a mechanism for propelling the comprehension system to keep up with the rapid pace of natural language.

DeLong et al.'s study has had an immense impact on the field of psycholinguistics, neurolinguistics and beyond. It is cited by authoritative reviews (e.g. *Altmann and Mirković, 2009*; *Hagoort, 2017*; *Lau et al., 2008*; *Pickering and Clark, 2014*; *Pickering and Garrod, 2007*) as delivering decisive evidence for probabilistic prediction of words all the way up to their phonological form. Moreover, as a demonstration of pre-activation of phonological form (sound) during reading, it is sometimes cited as evidence for 'prediction through production' (e.g. *Pickering and Garrod, 2013*), the hypothesis that linguistic predictions are implicitly generated by the language production system. To date, DeLong et al. has received a total of 766 citations (Google Scholar), averaging to more than one citation per week over the past decade, with an increasing number of citations in each subsequent year. The results also played an important role in settling an ongoing debate in the neuroscience of language. It provided the clearest evidence that the N400 component, which some researchers had long taken to directly index the high-level compositional processes by which people integrate a word's meaning with its context (*Brown and Hagoort, 1993*; *Chwilla et al., 1995*; *Connolly and Phillips, 1994*; *Friederici et al., 1999*; *van Berkum et al., 1999*; *Van Petten et al., 1999*), actually reflected non-compositional processes by which word information is accessed as a function of context (e.g. *Kutas and Hillyard, 1984*).

But how robust are gradient effects of form prediction? In over a decade that has passed since the publication by DeLong and colleagues, there is still no published study that directly replicates their graded pattern of results (for an overview, see *Ito et al., 2017b*). DeLong and colleagues also performed an alternative analysis of the same data, using cloze as a categorical variable instead of a continuous variable. This analysis did not yield a statistically significant result (p.59 in *DeLong, 2009*) and was not mentioned in the published report. In at least three other unpublished data sets (*DeLong, 2009*; *Miyamoto, 2016*), DeLong and colleagues did not find a significant correlation between article-N400 and cloze probability. *Martin et al. (2013)* claimed a successful conceptual replication in native speakers of English but not in bilinguals. However, their study did not test for a graded effect of cloze, and differed from the original in many crucial aspects of the experimental design, data-preprocessing and statistical analysis, clouding both a qualitative and quantitative comparison to the original results. Moreover, two attempts to replicate the Martin et al. results in English monolinguals failed to yield a reliable effect of cloze on article-ERPs (*Ito et al., 2017b*); for results that combined data from monolinguals and bilinguals, see *Ito et al., 2017a*).

As the tremendous scientific impact of the DeLong et al. findings is at odds with the apparent lack of replication attempts, we report here a direct replication study. Inspired by recent demonstrations for the need for large subject-samples in psychology and neuroscience research (*Button et al., 2013*; *Open Science Collaboration, 2015*), our replication spanned nine laboratories each with a sample size equal to or greater than that of the original. In addition to duplicating the original analysis, our replication attempt also seeks to improve upon DeLong et al.'s data analysis. DeLong et al.'s original analysis reduced an initial pool of 2560 data points (32 subjects who each read 80 sentences) to 10 grand-average values, by averaging N400 responses over trials within 10 cloze probability decile-bins (cloze 0–10, 11–20, etc.), per participant and then averaging over participants, even though these bins held greatly different numbers of observations (for example, the 0–10 cloze bin contained 37.5% of all data, whereas the 90–100 cloze bin contained only about 4%, which means that the reliability of the estimates per bin greatly differ, increasing the likelihood of obtaining spurious results; for additional discussion see *Ito et al., 2017b*). These 10 values were correlated with the average cloze value per bin, yielding numerically high correlation coefficients with large confidence intervals (e.g., the Cz electrode showed a statistically significant *r*-value of 0.68 with a 95% confidence interval ranging from 0.09 to 0.92). However, this analysis potentially compromises power by discretizing

cloze probability into deciles and not distinguishing various sources of subject-, item-, bin-, and trial-level variation. Furthermore, treating subjects as a fixed rather than random factor potentially inflates false positive rates, since the overall cloze effect is confounded with by-subject variation in the effect (*Barr et al., 2013*; *Clark, 1973*).

In our replication study, we followed two pre-registered analysis routes: a *replication analysis* that duplicated the DeLong et al. analysis, and a *single-trial analysis* that modelled variance at the level of item and subject (with a linear mixed-effects model), which offers better control over false-positives than the replication analysis when analyzing effects of the continuous predictor cloze probability. The effect of cloze on noun-elicited N400s (DeLong et al., 2015; *Kutas and Hillyard, 1984*) is necessary but not sufficient evidence for the claim on pre-activation in language processing (as it is also compatible with the view that the noun's cloze probability correlates with the ease of integration of that noun into the context). It serves as a manipulation check to ensure that the experiment is able to successfully detect graded variation in N400 amplitude, but does not provide strong evidence for the prediction of phonological form. That evidence would come from the ERPs elicited by articles. Observing a reliable effect of cloze on article-elicited N400s in the replication analysis and, in particular, in the single-trial analysis, would constitute powerful evidence for the pre-activation of phonological form during reading.

## Results

We first obtained offline cloze probabilities for all target articles and nouns from a group of native English speakers. These values closely resembled those of the original study (see Methods for details). In the subsequent ERP experiment, a different group of participants ($N$ = 334) read the sentences word-by-word from a computer display at a rate of 2 words per second, while we recorded their electrical brain activity at the scalp. The replication analysis and single-trial analysis described below were each pre-registered at https://osf.io/eyzaq/.

### Replication analysis

We sorted the articles and nouns into 10 bins based on each word's cloze probability (e.g. items with 0–10% cloze were put in one bin, 10–20% in another, etc.). For each laboratory, we averaged ERPs per bin first within, then across, participants. No baseline correction was used, following the procedure described in the Methods section in *DeLong et al. (2005)*. We then correlated the averaged cloze values per bin with mean ERP amplitude in the N400 time window (200–500 ms) elicited by the nouns (for the noun analysis) or articles (for the article analysis) from the corresponding bin, yielding a Pearson correlation coefficient (*r*-value) per EEG channel. This analysis yielded a very different pattern than DeLong et al. observed (*Figure 1*). In no laboratory did article-N400 amplitude at centro-parietal sites become significantly smaller (less negative) as article-cloze probability increased (in fact, in most laboratories the pattern went into the opposite direction). Only in one laboratory (Lab 2) did the correlation coefficient have a *p*-value below .05 in the predicted direction (positive) at any electrode (uncorrected for multiple comparisons), but this effect was observed at a few left-frontal electrodes, not at the central-parietal electrodes where DeLong et al. found their N400 effects. Moreover, in two laboratories (Labs 3 and 5), a statistically significant effect was observed in the opposite direction, larger (more negative) article-N400 amplitude for articles with increasing cloze probability. For the nouns, the pattern was more similar to the DeLong et al. results. In six laboratories (Lab 2, 3, 4, 6, 7, and 9), noun-N400 amplitude for nouns at central-parietal or parietal-occipital electrodes became smaller with increasing noun-cloze, and in two other laboratories (Lab 5 and 8) the effects clearly went in the expected direction without reaching statistical significance.

DeLong et al. recently mentioned using a 500 ms baseline correction procedure that was not mentioned in the published study (personal communication by DeLong, March 2017). In an exploratory analysis, we therefore recomputed the correlations based on data pooled from all laboratories using this baseline correction procedure (*Figure 2*). This analysis also showed a lack of statistically significant positive correlations for the articles, but statistically significant positive correlations for the nouns. In exploratory Bayesian analyses reported below, we perform an analysis to establish whether these results are consistent with the size and direction of the effects reported by DeLong et al., regardless of statistical significance.

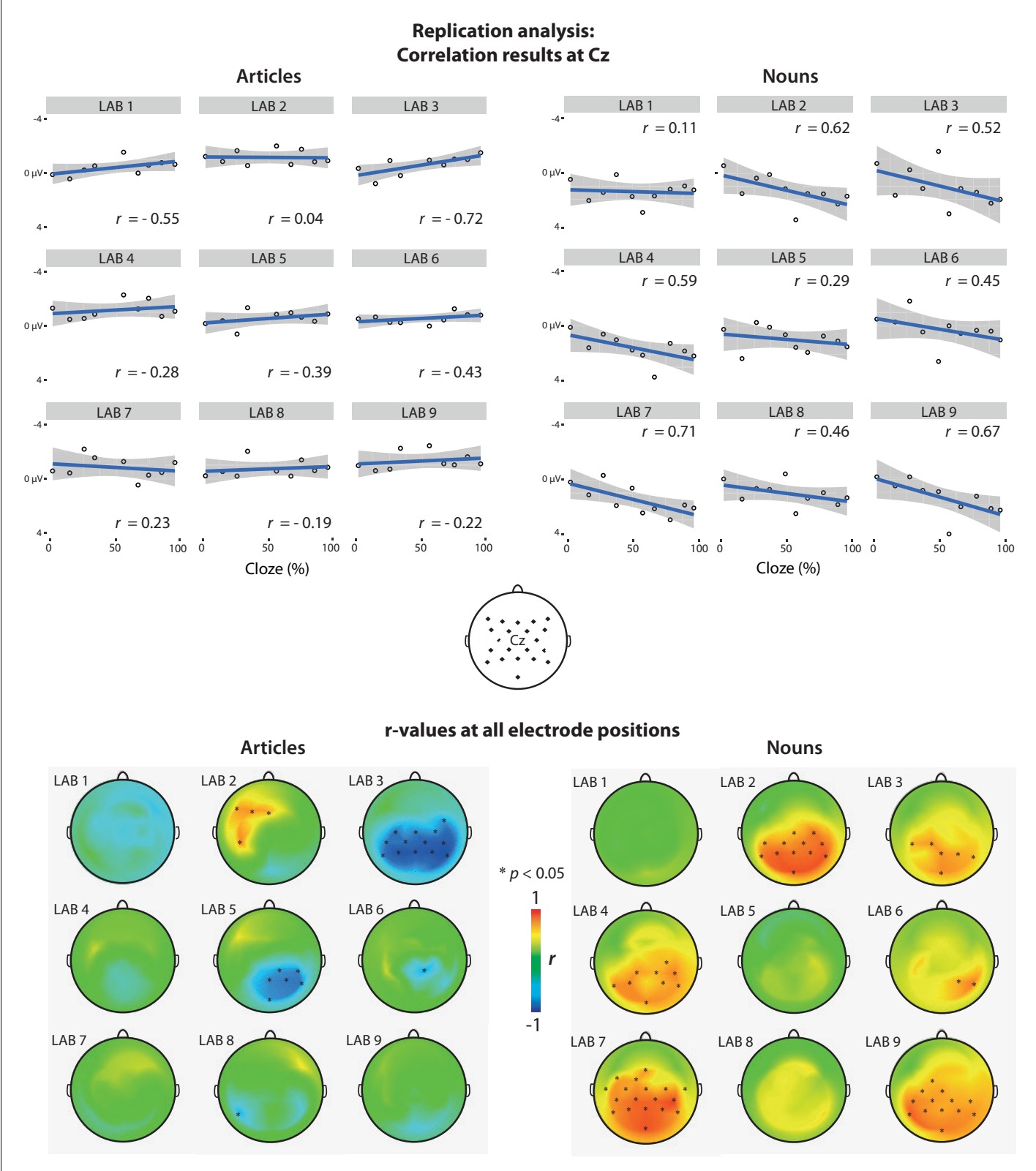

**Figure 1.** Replication analysis. Correlations between N400 amplitude and article/noun cloze probability per laboratory. N400 amplitude is the mean voltage in the 200–500 ms time window after word onset. A positive value corresponds to the canonical finding that N400 amplitude became smaller (less negative—more positive) with increasing cloze probability. Here and in all further plots, negative voltages are plotted upwards. Upper graph: Scatter plots showing the correlation between cloze and N400 activity at electrode Cz, for each lab. The position of Cz and the other electrodes is

*Figure 1 continued on next page*

*Figure 1 continued*

displayed in the head plot in between the upper and lower graph. Lower graph: Scalp distribution of the *r*-values for each lab. Asterisks (*) indicate electrodes that showed a statistically significant correlation (two-tailed p<0.05, not corrected for multiple comparisons). Exact *r*- and p-values for each laboratory and EEG channel are available as source data (*Figure 1—source datas 1–4*) and on https://osf.io/eyzaq.

DOI: https://doi.org/10.7554/eLife.33468.002

The following source data is available for figure 1:

**Source data 1.** *r*-values for the articles for each laboratory and each channel
DOI: https://doi.org/10.7554/eLife.33468.003
**Source data 2.** p-values for the articles for each laboratory and each channel.
DOI: https://doi.org/10.7554/eLife.33468.004
**Source data 3.** *r*-values for the nouns for each laboratory and each channel.
DOI: https://doi.org/10.7554/eLife.33468.005
**Source data 4.** *r*-values for the nouns for each laboratory and each channel.
DOI: https://doi.org/10.7554/eLife.33468.006

## Single-trial analysis

We first performed baseline correction by subtracting the average amplitude in the 100 ms time window before word onset. Baseline-corrected ERPs for relatively expected and unexpected words and difference waveforms are shown in *Figure 3*. Then, for the data pooled across all laboratories, we used linear mixed effects models to regress the N400 amplitude (in a spatiotemporal region of

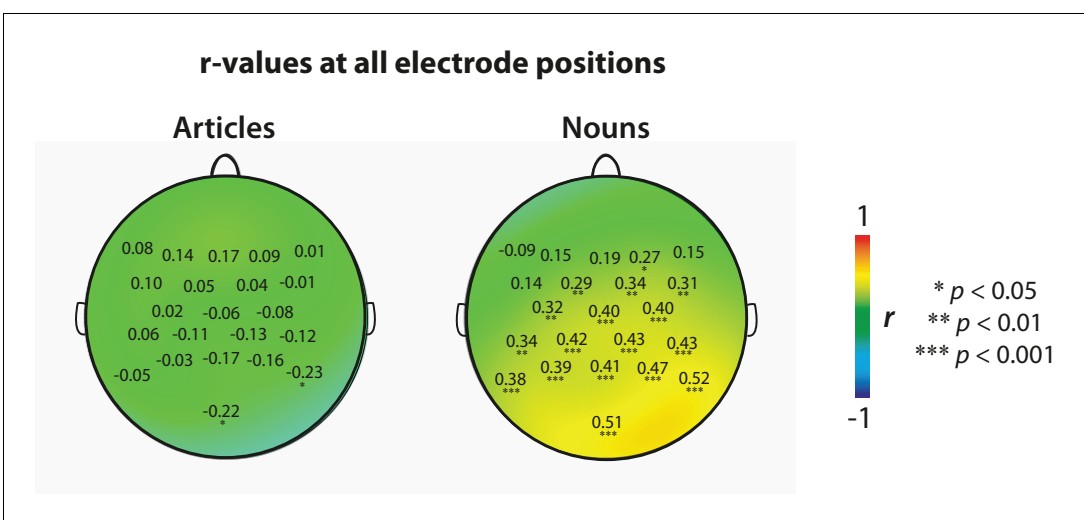

**Figure 2.** Replication analysis. Scalp distribution and *r*-values at each channel based on data pooled from all laboratories, using a 500 ms baseline correction procedure as used by *DeLong et al. (2005)*. Data were pooled after computing bin-averages per laboratory as in the original study, treating the laboratories as multiple observations of each bin-average. Asterisks (*) indicate electrodes that showed a statistically significant correlation (two-tailed, not corrected for multiple comparisons). Exact *r*- and p-values for each EEG channel are available as source data (*Figure 2—source datas 1–4*).
DOI: https://doi.org/10.7554/eLife.33468.007
The following source data is available for figure 2:

**Source data 1.** *r*-values for the articles for each channel, computed across laboratories.
DOI: https://doi.org/10.7554/eLife.33468.008
**Source data 2.** p-values for the articles for each channel, computed across laboratories.
DOI: https://doi.org/10.7554/eLife.33468.009
**Source data 3.** *r*-values for the nouns for each channel, computed across laboratories.
DOI: https://doi.org/10.7554/eLife.33468.010
**Source data 4.** p-values for the nouns for each channel, computed across laboratories.
DOI: https://doi.org/10.7554/eLife.33468.011

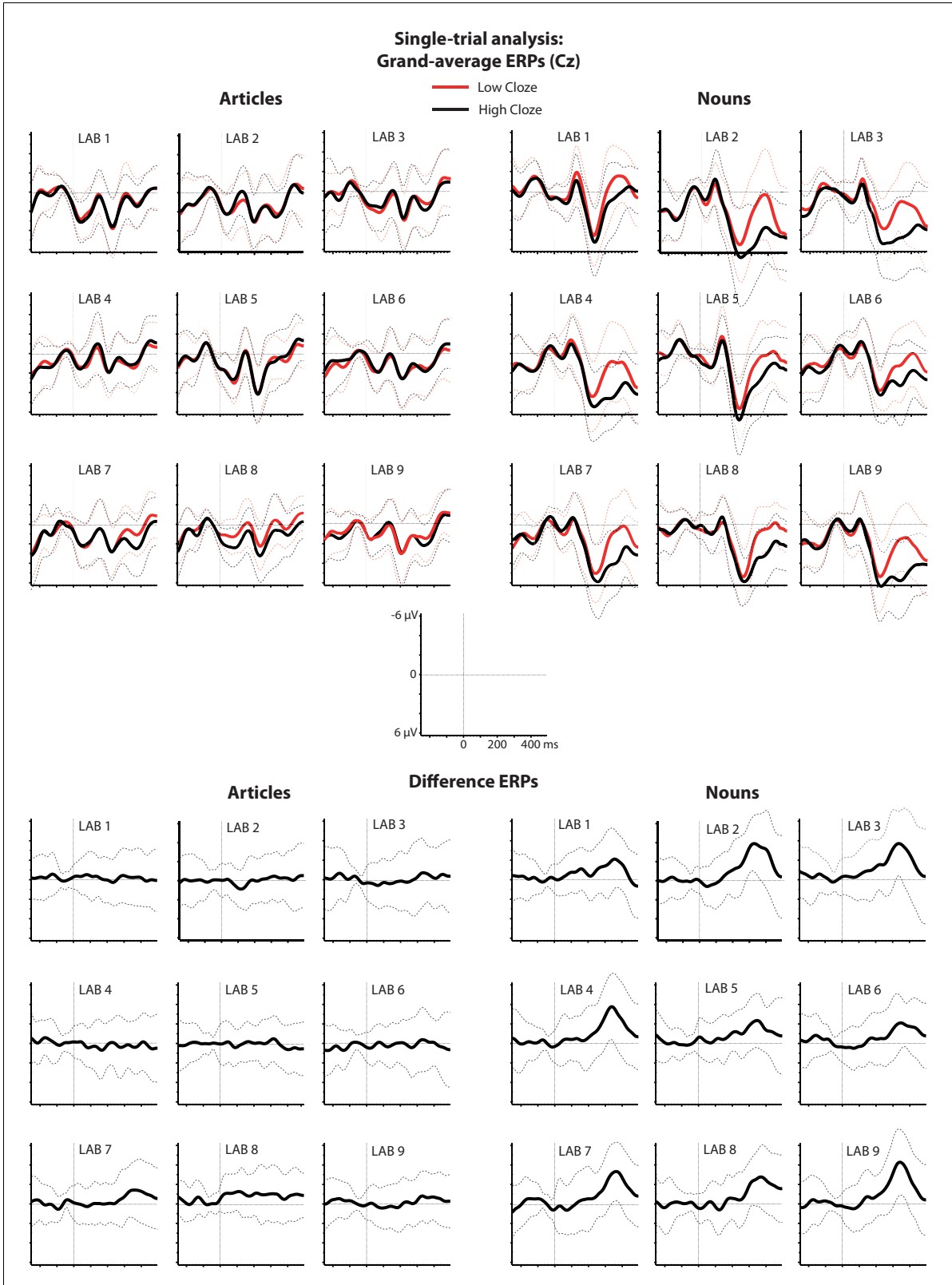

**Figure 3.** Single-trial analysis. Grand-average ERPs elicited by relatively expected and unexpected words (cloze higher/lower than 50%) and the associated difference waveforms (low minus high cloze) at electrode Cz. Dotted lines indicate one standard deviation above or below the grand average.

DOI: https://doi.org/10.7554/eLife.33468.012

interest selected *a priori* based on the DeLong et al. results) on cloze probability. For the articles, the effect of cloze was not statistically significant at the α = 0.05 level, $\beta$ = 0.29, CI [−0.08, .67], $\chi^2(1)$ =2.31, p=0.13 (see *Figure 4*, left panel) , with $\beta$ referring to the N400 difference in microvolts associated with stepping from 0% to 100% cloze. Unless otherwise indicated, p-values are two-tailed, and CIs are two-tailed 95% confidence intervals. The effect of cloze on N400 amplitude at the article did not significantly differ between laboratories, $\chi^2(8)$=7.90, p=0.44. For the nouns, however, higher cloze values were strongly associated with smaller N400s, $\beta$ = 2.22, CI [1.76, 2.69], $\chi^2(1)$=56.50, p<0.001 (see *Figure 4*, right panel). This pattern did not significantly differ between laboratories, $\chi^2(8)$=11.59, p=0.17. The effect of cloze on noun-N400s was statistically different from its effect on article-N400s, $\chi^2(1)$=31.38, p<0.001.

## Exploratory (i.e. not pre-registered) single-trial analyses

The effect of article-cloze did not significantly vary as a function of subject comprehension question accuracy, $\chi^2(1)$=0.45, p=0.50. In addition, the effect of article-cloze was also not statistically significant when subject comprehension accuracy was included in the analysis (100 ms baseline: $\beta$ = 0.24, CI [−0.17, .64], $\chi^2(1)$=1.27, p=0.26).

In our dataset, an analysis in the 500 to 100 ms time window *before* article-onset revealed a non-significant effect of cloze that resembled the pattern observed *after* article-onset, $\beta$ = 0.16, CI [−0.07, .39], $\chi^2(1)$=1.82, p=0.18 (*Figure 5*). Because the sentence context of each item was identical for the expected and unexpected article, effects in the pre-article window cannot be meaningfully related to the appearance of the article. Effects in this window must therefore be due to a spurious mix of 'residual EEG background noise' (activity that differed between expected and unexpected conditions but was unrelated to actual expectancy) with EEG activity associated with the specific word appearing before the article (which varied between items in terms of lexical characteristics, contextual constraint, and sentence position). The observed result in this time window therefore suggests that a 500 ms baseline correction procedure, which was used but not reported in *DeLong et al. (2005)*, would better correct for pre-article voltage-levels. We repeated our analysis with the 500 ms baseline correction procedure. Compared to the article-cloze effect observed in the

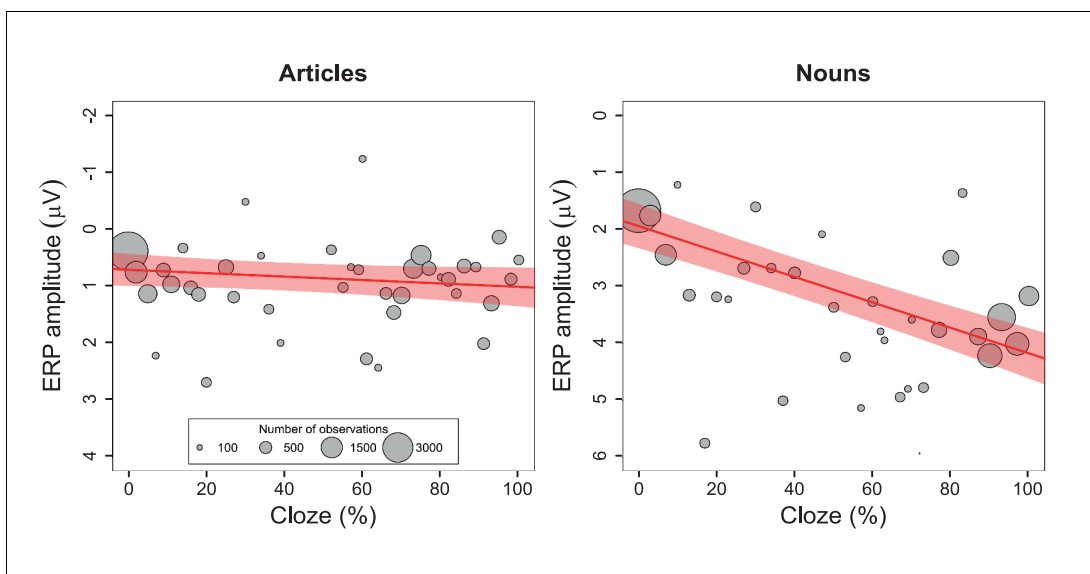

**Figure 4.** Single-trial analysis. Relationship between cloze and ERP amplitude for articles and nouns in the N400 spatiotemporal window, as illustrated by the mean ERP values per cloze value (number of observations reflected in circle size), along with the regression line and 95% confidence interval. A change in article cloze from 0 to 100 is associated with a change in amplitude of 0.296 μV (95% confidence interval: −0.08 to .67). A change in noun-cloze from 0 to 100 is associated with a change in amplitude of 2.22 μV (95% confidence interval: 1.75 to 2.69). The data for these analyses were pooled across all nine labs.
DOI: https://doi.org/10.7554/eLife.33468.013

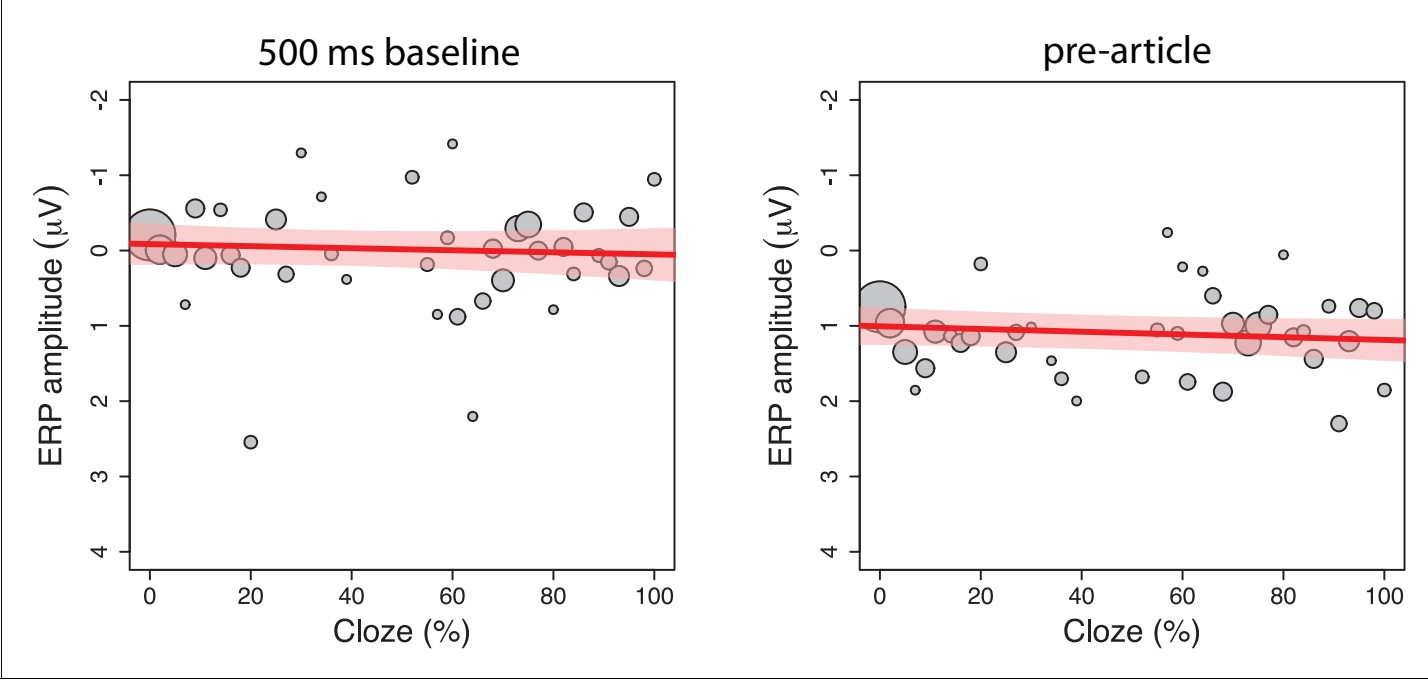

**Figure 5.** Exploratory single-trial analyses. The relationship between cloze and ERP amplitude as illustrated by the mean ERP values per cloze value (number of observations reflected in circle size), along with the regression line and 95% confidence interval, from two exploratory analyses. We performed a test which used a longer baseline time window (500 ms, left panel) to better control for pre-article voltage levels. This test reduced the initially observed effect of article-cloze, $\beta$ = 0.14, CI [−0.25, .53], $\chi^2(1)$=0.46, p=0.50. An analysis in the 500 to 100 ms time window *before* article-onset (right panel) revealed a non-significant effect of cloze that resembled the pattern observed *after* article-onset, $\beta$ = 0.16, CI [−0.07, .39], $\chi^2(1)$=1.82, p=0.18, shedding doubt on the conclusion that the observed results are due to the presentation of the articles.
DOI: https://doi.org/10.7554/eLife.33468.014

pre-registered analysis, the observed effect with the new baseline procedure (*Figure 5*) was numerically smaller and yielded a higher p-value ($\beta$ = 0.14, CI [−0.25, .53], $\chi^2(1)$=0.46, p=0.50).

Upon request of reviewers for this journal, we also performed an additional exploratory analysis with cloze as a dichotomous variable (based on a medium-split, thus disregarding the known variability in cloze values). We note that this type of analysis was not reported in *DeLong et al. (2005)*, although it was reported in the corresponding thesis chapter (*DeLong, 2009*) and did not yield a statistically significant effect of cloze on article-elicited ERPs. We performed this analysis for articles (100 and 500 ms baseline correction) and nouns. The results did not change substantially, and, in fact, each analysis yielded a lower $\chi^2$ value (and higher p-value) for the cloze variable than the corresponding analysis with cloze as a continuous predictor. The results can be reproduced from our online dataset and code.

## Exploratory Bayesian analyses

For the articles, our pre-registered replication analyses yielded non-significant p-values, indicating failure to reject the null-hypothesis that cloze has no effect on N400 activity. To better adjudicate between the null-hypothesis ($H_0$) and an alternative hypothesis ($H_r$), we performed an exploratory replication Bayes factor analysis for correlations (*Wagenmakers et al., 2016*). The obtained replication Bayes factor quantifies the evidence that there is an effect in the size and direction reported by DeLong et al. (see *Figure 6*). For the articles, this yielded strong to extremely strong evidence for the null hypothesis that the effect of cloze is zero, with $BF_{0r}$ values up to 154 (at the Cz electrode depicted by DeLong et al., $BF_{0r}$ = 77), and strongest evidence at the posterior channels. For the nouns, we obtained extremely strong evidence for the alternative hypothesis that the effect is consistent with the original effect, particularly at posterior channels, with $BF_{10}$ values up to 9,163,515 (at Cz, $BF_{r0}$ = 10,725). The pattern of results was similar when the 500 ms pre-stimulus baseline correction was applied.

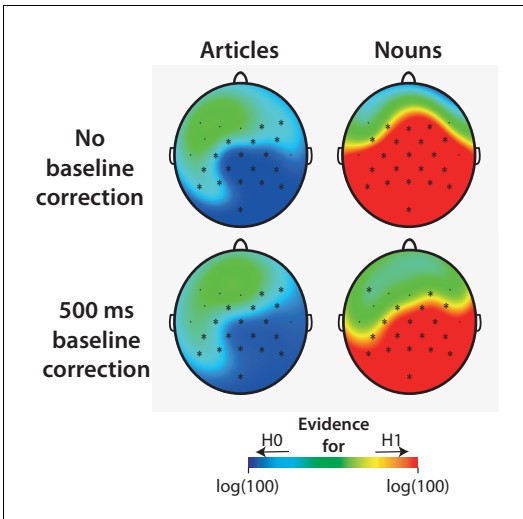

**Figure 6.** Exploratory replication Bayes factor analysis. This analysis quantifies the obtained evidence for the null hypothesis ($H_0$) that N400 is not impacted by cloze, or for the alternative hypothesis ($H_1$) that N400 is impacted by cloze with the direction *and* size of effect reported by DeLong et al. Scalp maps show the common logarithm of the replication Bayes factor for each electrode, capped at log(100) for presentation purposes. Electrodes that yielded at least moderate evidence for or against the null hypothesis (Bayes factor of ≥3) are marked by an asterisk. At posterior electrodes where DeLong et al. found their effects, our article data yielded strong to extremely strong evidence for the null hypothesis, whereas our noun data yielded extremely strong evidence for the alternative hypothesis (upper graphs). These results were obtained with the procedure described in DeLong et al. (no baseline correction), and with a 500 ms pre-word baseline correction (lower graphs), the procedure later described by DeLong and colleagues.
DOI: https://doi.org/10.7554/eLife.33468.015

Next, we computed Bayesian mixed-effect model estimates ($\beta$) and 95% credible intervals (CrI) for our single-trial analyses, using priors based on the results from DeLong et al. In both of our article-analyses, credible intervals included zero (100 ms baseline: $\beta = 0.31$, CrI [−0.06 .69]; 500 ms baseline: $\beta = 0.17$, CrI [−0.22 .55]). For the nouns, zero was not within the credible interval: $\beta = 2.24$, CrI [1.77 2.70]. The analyses suggest that the data (combined with prior assumptions about the effect) are not very consistent with the hypothesis that the article-effect is zero (further information and posterior summaries are available in *Figure 7*), but also are extremely inconsistent with the hypothesis that the article-effect is as big as that observed by *DeLong et al. (2005)*. The data are most consistent with an effect that is more likely to be positive than zero or negative but is very small (so small that it was not detected at traditional significance levels in this large-scale experiment with substantially higher power than previous experiments).

## Control experiment

Lack of a statistically significant, article-elicited prediction effect could reflect a general insensitivity of our participants to the phonologically conditioned variation of the English indefinite article, that is, *a/an* alternation. We ruled out this alternative explanation in an additional experiment that followed the replication experiment as part of the same experimental session. Participants read 80 short sentences containing the same nouns as the replication experiment, preceded by a phonologically licit or illicit article (e.g. 'David found a/an apple...''), presented in the same manner as before. In each laboratory, nouns following illicit articles elicited a late positive-going waveform compared to nouns following licit articles (see *Figure 8*), starting at about 500 ms after word onset and strongest at parietal electrodes. This standard P600 effect (*Osterhout and Holcomb, 1992*) was confirmed in a single-trial analysis, $\chi^2(1)=83.09$, $p<0.001$, and did not significantly differ between labs, $\chi^2(8)=8.98$, $p=0.35$.

## Discussion

In a landmark study, DeLong, Urbach and Kutas observed a statistically significant, graded modulation of article- and noun-elicited electrical brain potentials (N400) by the pre-determined probability that people continue a sentence fragment with that word (cloze). They concluded that people routinely and probabilistically pre-activate upcoming words to a high level of detail, including whether a word starts with a consonant or vowel. Our *direct replication* study spanning nine laboratories found a statistically significant effect of cloze on noun-elicited N400 activity but, critically, no significant effect of cloze on article-elicited N400 activity. This pattern was observed in a pre-registered replication analysis that duplicated the original study's analysis, and a pre-registered single-trial analysis that modeled variance at the level of item and subject. Exploratory replication Bayes factor analyses confirmed that we successfully replicated the direction and size of the correlations reported by

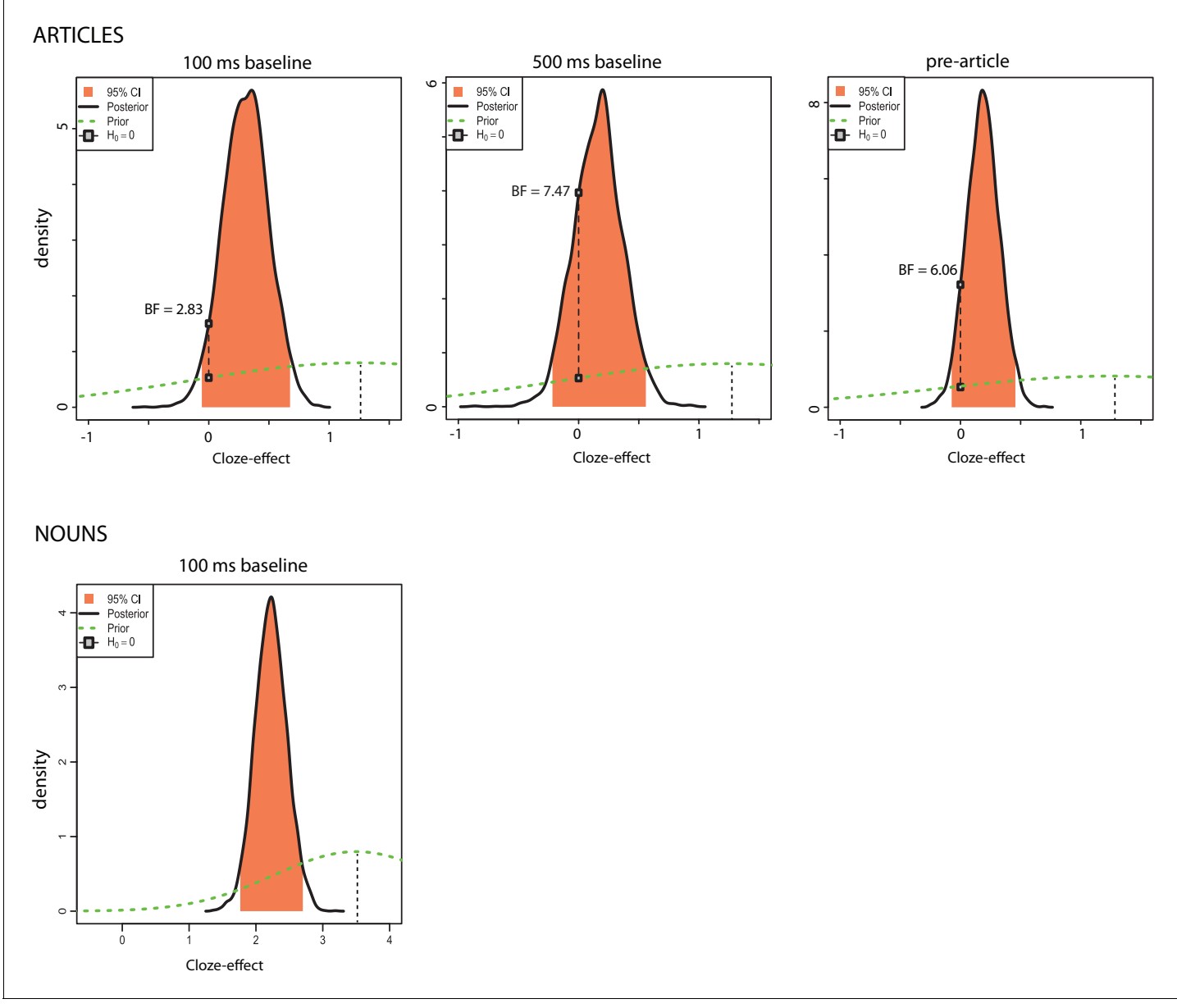

**Figure 7.** Exploratory Bayesian mixed-effects model analyses. Posterior density distributions for the effect of cloze on ERP amplitudes in the N400 window. The x-axis shows cloze effect sizes (i.e. changes in microvolts associated with an increase from 0% cloze probability to 100% cloze probability). The black line indicates the posterior distribution of effects; higher values of the posterior density at a given effect size indicate higher probability that this is the true effect size in the population. The peak of the posterior distribution roughly corresponds to the point estimate of the effect size (the regression coefficient) fitted from the Bayesian mixed effect model, i.e., the most likely value of the true effect size. The middle 95% of the posterior distribution, shaded in orange, corresponds to a two-tailed 95% credible interval for the effect size—i.e., an interval that we can be 95% confident contains the true effect. The green dotted line indicates the prior distribution (i.e., our expectation about where the true effect would lie before the data were collected). For the articles, this prior is centred on 1.25 μV, an approximation of the effect observed by *DeLong et al. (2005)*, and for the nouns it is centred on 3.5 μV. The black connected dots illustrate the ratio between the posterior and prior distribution (i.e. the Bayes factor) at the effect size of 0 μV; for example, a Bayes factor of 4 suggests we can be four times more certain that the true effect is zero after having conducted this experiment than before, or, in other words, that the data increased our confidence in the null effect of zero fourfold. We performed these analyses for each of the linear mixed-effects model analyses we performed. We note that in all the article-analyses, the posterior probability of the estimated effect being greater than zero is around 80 or 90%, although this is also true for the pre-stimulus variable, shedding doubt that the observed results are due to presentation of the articles. In none of our article-analyses did zero lie outside the obtained credible interval, whereas for the nouns, zero lay outside the credible interval. These results are consistent with a failure to replicate the size of the article-effect reported by DeLong et al. and a successful replication of the noun-effect.

DOI: https://doi.org/10.7554/eLife.33468.016

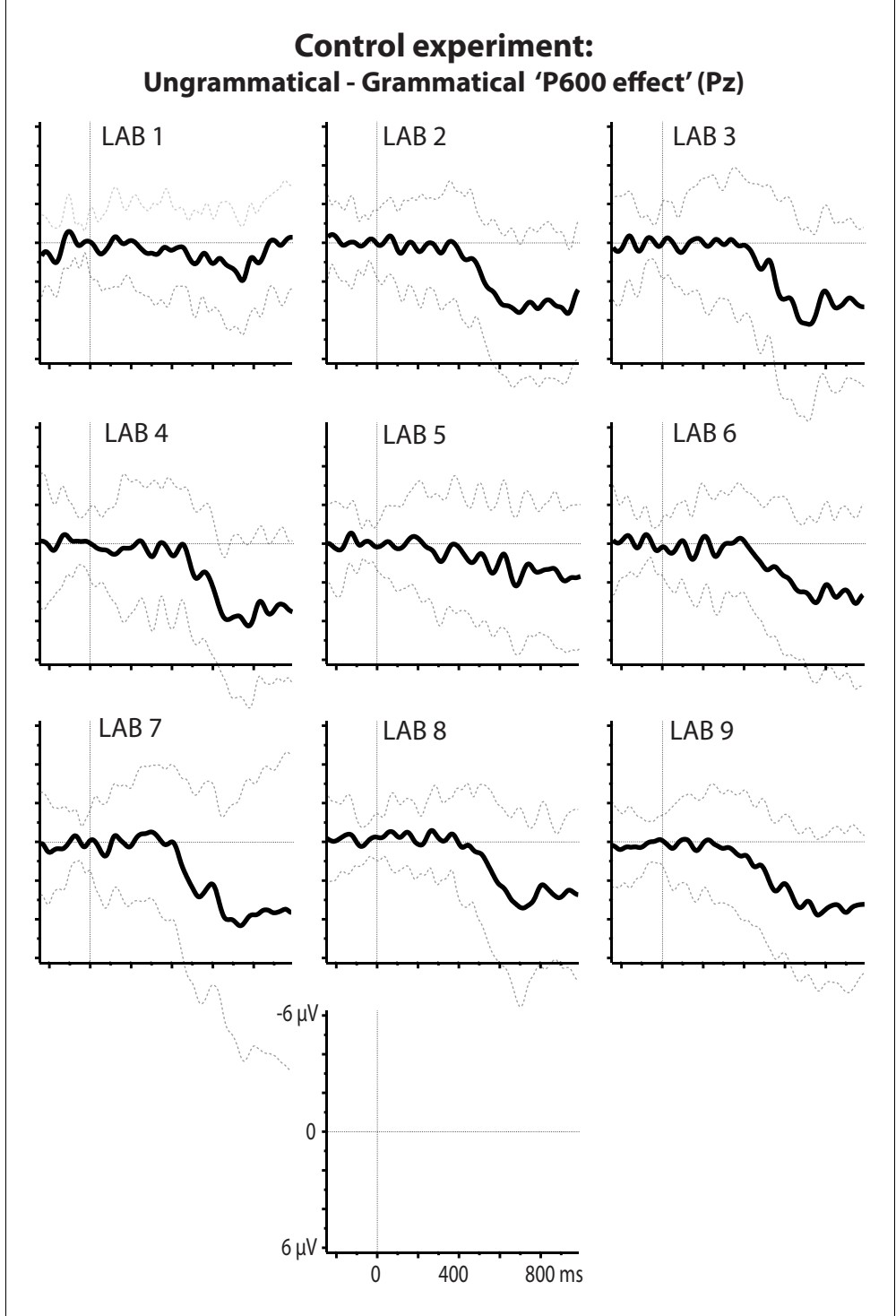

**Figure 8.** Control experiment. P600 effects at electrode Pz per lab associated with flouting of the English a/an rule. Plotted ERPs show the grand-average difference waveform and standard deviation for ERPs elicited by ungrammatical expressions ('an kite') minus those elicited by grammatical expressions ('a kite').
DOI: https://doi.org/10.7554/eLife.33468.017

Delong et al. for the nouns, but not for the articles. Exploratory Bayesian mixed-effects model analyses suggested that, while there is some evidence that the true population-level effect may be in the direction reported by DeLong and colleagues, the effect is likely far smaller than what they reported. In fact, the effect is likely is too small to be meaningfully observed without very large sample sizes. Finally, a control experiment confirmed that our participants did respect the phonological alternation *a/an* of the article with nouns used in the replication experiment.

Our findings thus challenge one empirical cornerstone of the 'strong prediction view' held by current theories of language comprehension (e.g. *Altmann and Mirković, 2009*; *Pickering and Garrod, 2013*). The strong prediction view entails two key claims. The first is that people pre-activate words at all levels of representation in a routine and implicit (i.e. non-strategic) fashion. Pre-activation is not limited to a word's meaning, but includes its grammatical features and even its orthographic and/or phonological form. This would put language on a par with other cognitive systems such as visual perception, wherein higher level brain regions attempt to predict lower level inputs (*Friston, 2005*; *2010*; *Summerfield and de Lange, 2014*). The second claim is that pre-activation occurs at all levels of contextual support and gradually increases in strength with the level of contextual support. When contextual support for a specific word is high, like at a 100% cloze value, the word's form and meaning is strongly pre-activated. When contextual support for a word is low, like when it is one amongst 20 words each with a 5% cloze value, pre-activation is distributed across multiple potential continuations. However, even then, a word's form and meaning are pre-activated, just weakly so. The strength of pre-activation is probabilistic, that is, linked to estimated probability of occurrence.

DeLong and colleagues, and subsequently other scientists (e.g. *Dell and Chang, 2014*; *Pickering and Garrod, 2013*), took their results as the evidence to support both these claims. *DeLong et al. (2005)* was – and still is - the only study to date that measured pre-activation at the prenominal articles *a* and *an* that do not differ in their semantic or grammatical content, and that observed a graded relationship between cloze and N400 activity across a range of low- and high-cloze words, rather than merely a difference between low- and high-cloze words. Given that the use of these articles depends on whether the next word starts with a vowel or consonant, their results were considered as powerful evidence that participants probabilistically pre-activated the initial sound of upcoming nouns.

However, we show that there is no statistically significant effect of cloze on article-elicited N400 activity, using a sample size more than ten times that of the original, and a statistical analysis that better accounts for sources of non-independence than the original averaging-based correlation approach. If an effect of cloze on article-N400s exists at all, its true effect size is so small that it cannot be reliably detected even in an expansive multi-laboratory approach, let alone in the typical sample size in psycholinguistic and neurolinguistic experiments (roughly, $N = 30$). This means that even if article-cloze is associated with a graded modulation of N400 amplitudes, this effect seems to be so small that it cannot be reliably measured with small samples, and thus the previous studies may not have contributed much reliable information to our understanding of this effect. Moreover, it is also possible that the effect is sensitive to specifics of the experimental procedure and context such that it lacks generalizability. Current theoretical positions thus either require new strong evidence for phonological pre-activation or require revision. In particular, one claim from the strong prediction view, namely that pre-activation routinely occurs across all – including phonological – levels (*Pickering and Garrod, 2013*), can no longer be viewed as having strong empirical support. Our work impels the field to think differently about what constitutes strong evidence within a theory, but also highlights the need for a theory of linguistic prediction to formulate quantitative predictions about the effect-size of to-be-observed effects (for discussion, see also *Vasishth et al., 2018*).

By contrast, we observed a strong and statistically significant effect of cloze on noun-elicited activity in the majority of our analyses. Although three of the nine laboratories did not show statistically significant correlations between noun-cloze and N400s, data pooled across all laboratories showed a strong and statistically significant noun-cloze effect, and our replication Bayes factor analysis overwhelmingly replicated the direction and size of the noun-cloze effect of DeLong et al. Moreover, our single-trial analysis revealed a significant noun-cloze effect in each of the laboratories, further demonstrating that our single-trial analysis is a more powerful approach than the averaged-based correlation approach of DeLong et al. These results are therefore consistent with the handful of studies that reported a graded relationship between noun-cloze and noun-N400s (*DeLong et al., 2005*; *Kutas and Hillyard, 1984*; *Wlotko and Federmeier, 2012*).

Where do our results leave the strong prediction view? Following the experimental logic of DeLong et al, we do not have sufficient evidence to conclude that people routinely pre-activate the initial phoneme of an upcoming noun, or perhaps any other word form information. Without pre-activation of the initial phoneme, the specific instantiation of the article does not cause people to revise their prediction about the meaning of the upcoming noun, thus lacking any impact on processing. Crucially, this conclusion is incompatible with the strong prediction view, because it suggests that pre-activation does not occur at the level of detail that is often assumed. Our results are also incompatible with an alternative interpretation of the DeLong et al. findings that people predict the article itself together with the noun (*Ito et al., 2017a*; *Van Petten and Luka, 2012*), and they pose a serious challenge to the theory that comprehenders predict upcoming words, including their initial phonemes, through implicit production (*Pickering and Garrod, 2013*). Crucially, the idea that prediction is probabilistic, rather than all-or-none, is now questionable, given that there is no other published report of a pre-activation gradient (also, see *Van Petten and Luka, 2012*, for a critique of the DeLong et al. conclusions that graded effects evidence graded pre-activation). Although other studies have claimed prediction of form (*Ito et al., 2016*) or a prediction gradient (*Smith and Levy, 2013*), no study has indisputably demonstrated graded pre-activation, that is, graded effects occurring *before* the noun. Effects that are observed upon, rather than before the noun, do not purely index pre-activation but can index a mixture of memory retrieval and semantic integration processes instigated by the noun itself (*Baggio and Hagoort, 2011*; *Lau et al., 2016*; Nieuwland et al., 2018; *Otten and Van Berkum, 2008*; *Steinhauer et al., 2017*). Therefore, there is currently no clear evidence to support routine probabilistic pre-activation of a noun's phonological form during sentence comprehension.

Our results, however, should not be taken as evidence against prediction in language processing more generally, and we believe that prediction could play an important role in language comprehension. In addition, our results do not necessarily exclude phonological form pre-activation, and we temper our conclusion with a caveat stemming from the *a/an* manipulation. For this manipulation to 'work', people must specifically predict the initial phoneme of the next word, and revise this prediction when faced with an unexpected article. However, because articles are only diagnostic about the next word within the noun phrase, rather than about the head noun itself, an unexpected article does not refute the upcoming noun, it merely signals that another word would come first (e.g., 'an old kite'). This opens up explanations for why the a/an manipulation 'fails' (see also *Ito et al., 2017a*, *2017b*). In addition, comprehenders may not predict the noun to follow immediately, but at a later point; the unexpected article then does not evoke a change in prediction. Predictions about a specific position may be disconfirmed too often in natural language to be viable. This idea is supported by corpus data (Corpus of Contemporary American English and British National Corpus), showing a mere 33% probability that a/an is directly followed by a noun. Alternatively, people predict the noun to come next, but only revise their prediction about its linear position while retaining the prediction about its meaning. So perhaps a revision of the predicted meaning, not the position, is required to trigger differential ERPs. In both of these hypothetical scenarios, people do not revise their prediction about the upcoming noun's meaning unless they must.

Our results can be straightforwardly reconciled with effects reported for other pre-nominal manipulations, such as those of Dutch or Spanish article-gender (e.g. *Van Berkum et al., 2005*; *Otten and Van Berkum, 2008*; *Otten and Van Berkum, 2009*; *Wicha et al., 2004*). Unlike a/an articles, gender-marked articles can immediately disconfirm the noun, because article- and noun-gender agrees regardless of intervening words (e.g. the Spanish article 'el' heralds a masculine noun). Revising the prediction about the noun presumably results in a semantic processing cost, thereby modulating N400 activity (e.g. Kochari and Flecken, 2017; *Otten and Van Berkum, 2009*). Although gender-marked articles do not consistently incur the exact same type of effect (for a recent review, see Kochari & Flecken, 2017) and have only been observed at very high-cloze values, previous studies suggest that a noun's grammatical gender can be pre-activated along with its meaning. Compared to this gender-manipulation, DeLong et al.'s study based on the English a/an manipulation claimed a stronger version of the prediction view, namely that people predict which word comes next up to its phonological form *and*, make backwards prediction as to the phonological form of the preceding linguistic material even on the basis of probabilistic, graded information.

What do our results say about prediction during natural language processing? Like the conclusions by DeLong et al., ours are limited by the generalization from comprehension of single

sentences in a laboratory setting. On one hand, a rich conversational or story context may enhance predictions of upcoming words, and listeners may be more likely to pre-activate the phonological form of upcoming words than readers. On the other hand, our laboratory setting offered particularly good conditions for prediction of the next word's initial sound to occur. Each article was always immediately followed by a noun, unlike in natural language. Moreover, our word presentation rate was slow compared to natural reading rates, which may facilitate predictive processing (*Ito et al., 2016*; *Wlotko and Federmeier, 2015*). In natural reading, articles are hardly fixated and often skipped (e.g. O'Regan 1979). In short, arguments can be made both for and against phonological form prediction in natural language settings, and novel avenues of experimentation are needed to settle this issue.

DeLong and colleagues recently stated an omission in the description of their data analysis, that is, a baseline procedure was applied to the data but inadvertently omitted from the description (*DeLong et al., 2005*). We have shown that our conclusions hold regardless of the baseline procedure. In a recent commentary, *DeLong et al., 2017* also described filler-sentences in their experiment, which were omitted from their original report, and were neither provided nor mentioned to us by the authors upon our request for the stimuli. DeLong et al. used the existence of these filler-sentences to dismiss an alternative explanation of their original findings, namely that an unusual experimental context wherein every sentence contains an article-noun combination leads participants to strategically predict upcoming nouns. Following this logic, our results were obtained *despite* an experimental context that could inadvertently encourage strategic prediction (for demonstrations of experimental context boosting predictive processing, see *Brothers et al., 2017*; *Lau et al., 2013*). Therefore, the presence of fillers in their experiment versus absence in ours cannot straightforwardly explain the different results, and may even strengthen our conclusions.

Since becoming publicly available as a pre-print (*Nieuwland et al., 2017*), our study has been simultaneously criticized for being not a sufficiently direct replication (due to the differences in fillers and baseline procedure; *DeLong et al., 2017*; *Yan et al., 2017*) and for being a too direct replication (because we base our analysis on the same theoretical assumptions as the original study, rather than applying an ad-hoc transformation or different kind of analysis that might 'reveal' the effect; e.g. *Yan et al., 2017*). As an example of the latter, an unpublished commentary by *Yan et al., 2017* raises an interesting point that cloze probability should be log-transformed to better approximate their suggested index of probabilistic semantic prediction, the Bayesian surprise over the noun semantics upon encountering the article. Yan et al. describe a number of exploratory reanalyses of our single-trial data with the log-transform, and one of those exploratory analyses yields a small but statistically significant effect of article-cloze (p=0.015). Ultimately, however, their conclusion is not that different from ours, namely that there is some evidence in our data that the effect is non-zero. More importantly, their commentary demonstrates that our dataset, like any complex EEG dataset, can be analyzed in many different ways, which can lead to different outcomes. However, even if alternative analyses are well-motivated after the fact, the problem remains that they are contingent on the data, and the accompanying researcher degrees of freedom lead to a multiple comparison problem (e.g. *Gelman and Loken, 2013*; *Luck and Gaspelin, 2017*). We pre-registered our main analyses and none of these allowed us to conclude that the DeLong et al. study replicated. Yan et al. present an alternative analysis that is exploratory and that itself requires further replication. Moreover, their analysis also raises a novel set of important concerns. For example, log-transformation of cloze also boosts the effect in the pre-article time window (p=0.058), where there cannot be a meaningful effect, possibly because it amplifies 'noise' (between-item differences at the low end of the cloze-scale that have nothing to do with prediction of the article). Furthermore, log-transformation does not yield a significant effect with the original baseline procedure of DeLong et al., and it strongly boosts the impact of items with zero cloze, that is, the items that are problematic because their predictability cannot be accurately estimated (of note, without zero-cloze values in their analysis, higher cloze leads to more negative, not positive voltage). Yan et al. report that log-transformation yields somewhat higher *t*-values of cloze in this dataset and changes our non-significant effect into a significant effect, but it remains unclear whether log-transformation is indeed 'better'. Crucially, the difference between significant and not-significant itself may not be significant (*Gelman and Stern, 2006*), log-transformation does not yield higher t-values consistently across laboratories, does not necessarily improve model fit, and does not yield higher t-values or improve model fit in another large dataset (collapsed data from *Ito et al., 2017a*; *Nieuwland, 2016*;

*Nieuwland and Martin, 2012*; total N = 124). Finally, it is unknown whether log-transformation weakens rather than strengthens the effect of the original study. Details of these and further concerns are available on https://osf.io/mb2ud. In sum, these concerns merely add to our main point, namely that even if analysis decisions are justifiable in retrospect, a flexible analysis practice can lead researchers to capitalize on noise (*Gelman and Loken, 2013*).

To conclude, we failed to replicate the main result of DeLong et al., a landmark study published more than 10 years ago that has not been directly replicated since. Our results suggest that, if there is an effect of article-cloze probability on the amplitude of the N400, it is too small and/or too sensitive to unknown experimental design factors to have been meaningfully measured in previous small-sample-size experiments. Our findings thus do not lend clear support to the 'strong prediction view' in which people routinely and probabilistically pre-activate information at all levels of linguistic representation, including phonological form information such as the initial phoneme of an upcoming noun. Consequently, there is currently no convincing evidence that people routinely pre-activate the phonological form of an upcoming noun during written sentence comprehension. In addition, our findings further highlight the importance of direct replication, large sample size studies, transparent reporting and pre-registration to advance reproducibility and replicability in the neurosciences.

## Materials and methods

### Experimental design and materials

Nieuwland requested all original materials from DeLong et al., including the questions and norms, with the stated purpose of direct replication (personal communication, November 4 and 19, 2015), upon which DeLong et al. made available the 80 sentences described in the original study. These sentences were then adapted from American to British spelling and underwent a few minor changes to ensure their suitability for British participants. The complete set of materials and the list of changes to the original materials are available online (*Supplementary file 1*). The materials were 80 sentence contexts with two possible continuations each: a more or less expected indefinite article + noun combination. The noun was followed by at least one subsequent word. All article + noun continuations were grammatically correct. Each article + noun combination served once as the more expected continuation and the other time as the less expected continuation, in different contexts. We divided the 160 items in two lists of 80 sentences such that each list contained each noun only once. Each participant was presented with only one list (thus, each context was seen only once). One in four sentences was followed by a yes/no comprehension question, which yielded a mean response accuracy of 95% (after taking into account ambiguity in three of the questions, see *Supplementary file 1*). While this percentage is very similar to that reported by DeLong et al., we note that this cannot be directly compared to the accuracy reported in DeLong et al., because we had to create new comprehension questions in the absence of the original ones. Regardless, because Delong et al. suggested that our results were due to poor language comprehension (*DeLong et al., 2017*), we describe an exploratory analysis in which we attempt to account for variation in response accuracy in the statistical model.

We obtained article cloze and noun cloze ratings from a separate group of native speakers of English who were students at the University of Edinburgh and did not participate in the ERP experiment. They were instructed to complete the sentence fragment with the best continuation that comes to mind (*Taylor, 1953*). We obtained article cloze ratings from 44 participants for 80 sentence contexts truncated before the critical article. Noun cloze ratings were obtained by first truncating the sentences after the critical articles, and presenting two different, counterbalanced lists of 80 sentences to 30 participants each, such that a given participant only saw each sentence context with the expected or the unexpected article. The obtained values closely resemble those of the original study, with the same range (0–100% for articles and nouns), slightly lower median values (for articles and nouns, 29% and 40%, compared to 31% and 46% in the original study), but slightly higher mean values (for articles and nouns, 41% and 46%, compared to 36% and 44%). Because the sentence materials we used describe common situations that can be understood by any English speaker, and because students at the University of Edinburgh come from across the whole of the UK, we had no *a priori* expectation that cloze ratings would differ substantially across laboratories, and thus we did not obtain cloze norms from other sites. Consistent with this assumption, nothing in our results

suggests stronger cloze effects in University of Edinburgh students compared to other students, suggesting that our cloze norms are sufficiently representative for the other universities. The raw cloze responses are available on our OSF page.

## Participants

Participants were students from the University of Birmingham, Bristol, Edinburgh, Glasgow, Kent, Oxford, Stirling, York, or volunteers from the participant pool of University College London or Oxford University, who received cash or course credit for taking part in the ERP experiment. Participant information and EEG recording information per laboratory is available online (*Supplementary file 1*). We pre-registered a target sample size of 40 participants per laboratory, which was thought to give at least 32 participants (the sample size of DeLong et al.) per laboratory after accounting for data loss, as was later confirmed. Due to logistic constraints, not all laboratories reached an N of 40. Because in two labs corruption of data was incorrectly assumed before computing trial loss, these laboratories tested slightly more than 40 participants. All participants ($N$ = 356; 222 women) were right-handed, native English speakers with normal or corrected-to-normal vision, between 18 and 35 years (mean, 19.8 years), free from any known language or learning disorder. Eighty-nine participants reported a left-handed parent or sibling.

## Procedure

After giving written informed consent, participants were tested in a single session. Sentences were presented visually in the center of a computer display, one word at a time (200 ms duration, followed by a blank screen of 300 ms duration). Due to a programming error, in four labs (1, 3, 5 and 8, which used E-prime scripts) the critical articles and nouns, but not other words, were followed by a 380 ms blank instead of the intended 300 ms. This delay is unlikely to have affected the results because if it was noticed at all, which is unlikely, it could only be noticed 500 ms after the article, that is, after the N400 window associated with the article. Of note, the pattern of the results from the pre-registered single-trial analysis did not change when we removed these labs from the analysis. Participants were instructed to read sentences for comprehension and answer yes/no comprehension questions by pressing hand-held buttons. The electroencephalogram (EEG) was recorded from at least 32 electrodes.

The replication experiment was followed by a control experiment, which served to detect sensitivity to the correct use of the a/an rule in our participants. Participants read 80 relatively short sentences (average length eight words, range 5–11) that contained the same critical words as the replication experiment, preceded by a correct or incorrect article. As in the replication experiment, each critical word was presented only once, and was followed by at least one more word. All words were presented at the same rate as the replication experiment. There were no comprehension questions in this experiment. After the control experiment, participants performed a Verbal Fluency Test and a Reading Span test; the results from these tests are not discussed here. All stimulus presentation scripts are publicly available in two different software packages (E-Prime and Presentation) on https://osf.io/eyzaq.

## Data processing

Data processing was performed in BrainVision Analyzer 2.1 (Brain Products, Germany). We performed one pre-registered replication analysis that followed the DeLong et al. analysis as closely as possible and one pre-registered single-trial analysis (Open Science Framework, https://osf.io/eyzaq). All non-pre-registered analyses are considered as exploratory. First, we interpolated bad channels from surrounding channels, and downsampled to a common set of 22 EEG channels per laboratory which were similar in scalp location to those used by DeLong et al. One laboratory did not have 12 of the selected 22 channels in its EEG channel montage, and we matched the full 22-channel layout used for other laboratories by creating 12 virtual channels from neighbouring channels using topographic interpolation by spherical splines. We then applied a 0.01–100 Hz digital band-pass filter (including 50 Hz Notch filter), re-referenced all channels to the average of the left and right mastoid channels (in a few participants with a noisy mastoid channel, only one mastoid channel was used), and segmented the continuous data into epochs from 500 ms before to 1000 ms after word onset. We then performed visual inspection of all data segments and rejected data with amplifier blocking,

movement artifacts, or excessive muscle activity. Subsequently, we performed independent component analysis (*Jung et al., 2000*) on a 1 Hz high-pass filtered version of the data, and applied the obtained weightings to the original data to correct for blinks, eye movements or steady muscle artefacts. After this, we automatically rejected segments containing a voltage difference of over 120 μV in a time window of 150 ms or containing a voltage step of over 50 μV/ms. Participants with fewer than 60/80 article trials or 60/80 noun trials were removed from the analysis, leaving a total of 334 participants (range across laboratories 32–42, and therefore each lab had a sample size at least as large as DeLong et al.). On average, participants had 77 article trials and 77 noun trials. All raw data and pre-processed data are available on https://osf.io/eyzaq.

## Pre-registered replication analysis

We applied a 4th-order Butterworth band-pass filter at 0.2–15 Hz to the segmented data, averaged trials per participant within 10% cloze bins (0–10, 11–20, etc. until 91–100), and then averaged the participant-wise averages separately for each laboratory. Because the bins did not contain equal numbers of trials (the intermediate bins contained fewest trials), like in DeLong et al., not all participants contributed a value for each bin to the grand average per laboratory. For nouns and articles separately, and for each EEG channel, we computed the correlation between ERP amplitude in the 200–500 ms time window per bin with the average cloze probability per bin.

## Pre-registered single-trial analysis

In this analysis, we did not apply the 0.2–15 Hz band-pass filter, which carries the risk of inducing data distortions (Luck, 2014; *Tanner et al., 2015*). However, we deemed it necessary to perform a baseline correction of the data. This procedure corrects for spurious voltage differences before word onset, increasing confidence that observed effects are elicited by the word rather than differences in brain activity that already existed before the word and is a standard procedure in ERP research (Luck, 2014). *DeLong et al. (2005)* did not report a baseline correction, nor did any of the related work from DeLong and colleagues that was reported in *DeLong, 2009*. Yet baseline correction has been used in many other publications from the Kutas Cognitive Electrophysiology Lab. We chose a 100 ms pre-stimulus baseline as the most frequently used one both in other studies from the Kutas lab and in similar studies from other labs. For each trial, we performed baseline correction by subtracting the mean voltage of the −100 to 0 ms time window from each data point in the epoch.

Instead of averaging N400 data across trials and participants for subsequent statistical analysis, we performed linear mixed effects model analysis (*Baayen et al., 2008*) of the single-trial N400 data, using the 'lme4' package (Bates, Maechler, Bolker & Walker, 2014) in the R software (R Core-Team, 2014). This approach simultaneously models variance associated with each subject and with each item. Especially when analyzing effects of a continuous predictor variable such as cloze probability, linear mixed-effects regression offers better control over false-positive results than the averaged-based correlation analysis of the original study. Using a spatiotemporal region-of-interest approach based on the DeLong et al. results, our dependent measure (N400 amplitude) was the average voltage across six centro-parietal channels (Cz/C3/C4/Pz/P3/P4) in the 200–500 ms window for each trial. Analysis scripts and data to run these scripts are publicly available on https://osf.io/eyzaq.

For articles and nouns separately, we used a maximal random effects structure as justified by the design (*Barr et al., 2013*), which did not include random effects for 'laboratory' as there were only nine laboratories. Z-scored cloze was entered in the model as a continuous variable, and laboratory was entered as a deviation-coded nuisance predictor. We tested the effects of 'laboratory' and 'cloze' through model comparison with a $\chi^2$ log-likelihood test. We tested whether the inclusion of a given fixed effect led to a significantly better model fit. The first model comparison examined laboratory effects, namely whether the cloze effect varied across laboratories (cloze-by-laboratory interaction) or whether the N400 magnitudes varied over laboratory (laboratory main effect). Because laboratory effects were not significant, we dropped them from the analysis because they were not of theoretical interest. For the articles and nouns separately, we compared the subsequent models below. Each model included the random effects associated with the fixed effect 'cloze' (see *Barr et al., 2013*). All output $\beta$ estimates and 95% confidence intervals (CI) were transformed from

z-scores back to raw scores, and then back to the 0–100% cloze range, so that the voltage estimates represent the change in voltage associated with a change in cloze probability from 0 to 100.

 Model 1: N400 ~cloze * laboratory + (cloze | subject) + (cloze | item)
 Model 2: N400 ~cloze + laboratory + (cloze | subject) + (cloze | item)
 Model 3: N400 ~cloze + (cloze | subject) + (cloze | item)
 Model 4: N400 ~ (cloze | subject) + (cloze | item)

In an analysis that itself was not pre-registered but that included the data from the pre-registered analysis of both articles and nouns, we tested the differential effect of cloze on article ERPs and on noun ERPs by comparing models with and without an interaction between cloze and the deviation-coded factor 'wordtype' (article/noun). Random correlations were removed for the models to converge.

 Model 1: N400 ~cloze * wordtype + (cloze * wordtype || subject) + (cloze * wordtype || item)
 Model 2: N400 ~cloze + wordtype + (cloze * wordtype || subject) + (cloze * wordtype || item)

## Exploratory correlation analysis

Of note, DeLong et al. have recently described using a 500 ms baseline correction procedure that they failed to mention in *DeLong et al. (2005)*. Using this baseline correction procedure, we recomputed the correlations that we obtained in our Replication analysis (*Figure 2*). To compare our results most directly with those reported in *Figure 1C* of *DeLong et al. (2005)*, we pooled data from all the laboratories to obtain a single *r*-value for each EEG-channel. Data were pooled after computing bin-averages per laboratory as in the original study, treating the laboratories as multiple observations of each bin-average.

## Exploratory single-trial analyses

We performed an exploratory analysis in the 500 to 100 ms time window *before* the article, using the originally (−100 to 0 ms) baselined data, using Models 3 and 4 from the article analysis. This window covers the first 400 ms of the word that preceded the article. Analysis in this window yielded a similar pattern as in the pre-registered analysis, which indicates that a baseline correction procedure covering the entire 500 ms pre-stimulus window would account better for pre-article voltage levels. We performed this additional analysis, the results of which did not change our conclusions and are shown in *Figure 5*.

We also performed an exploratory analysis in which we control for a potential influence of response accuracy, taken as a proxy for the subject's attention to the task, on predictive processing of linguistic input. We entered the (z-transformed) average response accuracy of each subject in our model, and compared the models below. Comparison of Models 1 and 2 tested whether the effect of cloze on the article-N400s depended on subject accuracy. Comparison of Models 2 and 3 tested whether there was a significant effect of cloze on article-N400s when subject accuracy was included in the model.

 Model 1: N400 ~accuracy * cloze + (cloze | subject) + (cloze | item)
 Model 2: N400 ~accuracy + cloze + (cloze | subject) + (cloze | item)
 Model 3: N400 ~accuracy + (cloze | subject) + (cloze | item)

## Exploratory Bayesian analyses

Supplementing the Replication analysis, we performed a replication Bayes factor analysis for correlations (*Wagenmakers et al., 2016*) using as prior the size and direction of the effect reported in the original study. We performed this test for each electrode separately, after collapsing the data points from the different laboratories. Because we had no articles in the 40–50% cloze bin, there was a total of 9 and 10 data points per laboratory for the articles and nouns, respectively. Our analysis used priors estimated from the DeLong et al. results, matched as closely as possible to our electrode locations. A Bayes factor between 3 and 10 is considered moderate evidence, between 10 and 30 is considered strong evidence, 30–100 is very strong evidence, and values over 100 are considered extremely strong evidence (Jeffreys, 1961). In addition to using a 100 ms pre-stimulus baseline, we also computed the replication Bayes factors using the 500 ms pre-stimulus time window for baseline correction. Results are shown in Figure 6.

Supplementing the pre-registered single-trial analyses, we performed an exploratory Bayesian mixed-effects model analysis using the brms package for R (Buerkner, 2016), which fits Bayesian multilevel models using the Stan programming language (Stan Development Team, 2016). Nieuwland requested to use the results of a mixed-effects model reanalysis of the DeLong et al. data as an appropriate prior (personal communication from Nieuwland, November 14 and 22 2017); this request was declined by DeLong and colleagues. We were therefore limited to using a prior centered on a point estimate based on the Delong et al. correlation analysis, namely our estimate of the observed effect size at Cz for a difference between 0% cloze and 100% cloze (1.25 μV and 3.75 μV for articles and nouns, respectively, based on visual inspection of the graphs) and a prior centered on zero for the intercept. Both priors had a normal distribution and a standard deviation of 0.5 (given the a priori expectation that average ERP voltages in this window generally fluctuate on the order of a few microvolts; note that these units are expressed in terms of the z-scored cloze values, rather than the original cloze values, such that μ for the cloze prior was 0.45, which corresponds to a raw cloze effect of 1.25). We computed estimates and 95% credible intervals for each of the mixed-effects models we tested, and transformed these back into raw cloze units. The credible interval is the range of values such that one can be 95% certain that it contains the true effect, given the data, priors and the model. The results from these analyses are shown in *Figure 7*; the analyses suggest that, while there may be a small positive association between article cloze and ERP amplitude elicited by the articles, the effect is substantially smaller than that estimated by *DeLong et al. (2005)* and likely is too small to be observed without very large sample sizes.

## Control experiment

Analysis of the control experiment involved a comparison between a model with the categorical factor 'grammaticality' (grammatical/ungrammatical) and a model without. Our dependent measure (P600 amplitude; *Osterhout and Holcomb, 1992*; Wicha et al., 2004) was the average voltage across six centro-parietal channels (Cz/C3/C4/Pz/P3/P4) in the 500–800 ms window for each trial. Results are shown in *Figure 8*.

Model 1: P600 ~grammaticality + (grammaticality | subject) + (grammaticality | item)
Model 2: P600 ~ (grammaticality | subject) + (grammaticality | item)

## Acknowledgements

This work was partly funded by ERC Starting grant 636458 to HJF. We thank Matt Davis for his comments on a previous draft of this work. We thank Alexander Ly and Eric-Jan Wagenmakers for their support in computing the replication Bayes factors.

## Additional information

### Funding

| Funder | Grant reference number | Author |
|---|---|---|
| European Research Council | ERC Starting grant 636458 | Heather J Ferguson |

The funders had no role in study design, data collection and interpretation, or the decision to submit the work for publication.

### Author contributions

Mante S Nieuwland, Conceptualization, Data curation, Software, Formal analysis, Supervision, Validation, Investigation, Visualization, Methodology, Writing—original draft, Project administration, Writing—review and editing; Stephen Politzer-Ahles, Software, Formal analysis, Validation, Investigation, Visualization, Methodology, Writing—original draft, Project administration, Writing—review and editing; Evelien Heyselaar, Aine Ito, Investigation, Writing—review and editing; Katrien Segaert, E Matthew Husband, David I Donaldson, Resources, Supervision, Writing—review and editing; Emily Darley, Sarah Von Grebmer Zu Wolfsthurn, Federica Bartolozzi, Vita Kogan, Diane Mézière, Xiao Fu, Zdenko Kohút, Investigation; Nina Kazanina, Resources, Supervision, Writing—original draft, Writing—review and editing; Dale J Barr, Software, Formal analysis, Writing—review and editing;

Guillaume A Rousselet, Resources, Formal analysis, Supervision, Methodology, Writing—review and editing; Heather J Ferguson, Resources, Supervision, Funding acquisition, Writing—review and editing; Simon Busch-Moreno, Eugenia Kulakova, Software, Formal analysis, Investigation; Jyrki Tuomainen, Resources, Supervision; Shirley-Ann Rueschemeyer, Supervision, Writing—review and editing; Falk Huettig, Conceptualization, Funding acquisition, Writing—review and editing

**Author ORCIDs**
Mante S Nieuwland (iD) http://orcid.org/0000-0003-4001-6608
Stephen Politzer-Ahles (iD) http://orcid.org/0000-0002-5474-7930
Nina Kazanina (iD) http://orcid.org/0000-0001-7737-4279
Aine Ito (iD) http://orcid.org/0000-0003-4408-8801
Guillaume A Rousselet (iD) http://orcid.org/0000-0003-0006-8729

**Ethics**
Human subjects: All participants were informed about the procedure of the experiment and then gave informed consent to use the data for research and dissemination/publication purpose. Ethical approval for EEG experimentation was obtained at each involved institution, according to custom guidelines of the ethics committee at each institution.

**Decision letter and Author response**
Decision letter https://doi.org/10.7554/eLife.33468.023
Author response https://doi.org/10.7554/eLife.33468.024

## Additional files

### Supplementary files
• Supplementary file1. This file contains Supplementary Tables 1-3. Supplementary Table 1 contains the sentence materials with cloze probabilities (0-100%) of articles and nouns, along with post-noun sentence endings, comprehension questions and expected answer. Of note, because expectedness of the noun is here determined by the cloze value of the preceding article, there are three items (28, 29 and 49) in which the unexpected noun has a cloze that is equal to or higher than the expected noun. This has no repercussions for the statistical results because the noun-analysis was based on noun cloze. Supplementary Table 2 contains the List of changes to the materials used by Delong et al. (2005). Supplementary Table 3 contains detailed information about participants, trial numbers and EEG recording equipment per laboratory.
DOI: https://doi.org/10.7554/eLife.33468.018
• Transparent reporting form
DOI: https://doi.org/10.7554/eLife.33468.019

### Major datasets
The following dataset was generated:

| Author(s) | Year | Dataset title | Dataset URL | Database, license, and accessibility information |
|---|---|---|---|---|
| Nieuwland M | 2018 | Replication Recipe Analysis plan | https://osf.io/eyzaq/ | Available at the Open Science Framework |

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
