## [Decision Letter]

Thank you for submitting your article "Large-scale replication study reveals a limit on probabilistic prediction in language comprehension" for consideration by *eLife*. Your article has been reviewed by three peer reviewers, and the evaluation has been overseen by a Reviewing Editor and Sabine Kastner as the Senior Editor. The following individual involved in review of your submission have agreed to reveal his identity: Matt Davis (Reviewer #3).

The reviewers have discussed the reviews with one another and the Reviewing Editor has drafted this decision to help you prepare a revised submission. The three reviewers were unanimous in recognizing the importance of this large-scale replication (with pre-registered hypotheses, across multiple labs) study, and of publishing this kind of work. They all agreed that the study is appropriate for *eLife*, but also all agreed that the presentation requires some polishing before publication.

The positives here are clear. While no experimental evidence can "prove" the null hypothesis, this work highlights that the field has adopted a very strong assumption about mechanisms of language comprehension even though the landmark evidence for this idea is not very robust. The reviewers all applaud the effort that went into this large-scale, multi-site study. All hope that the publication of this study encourages others to undertake this kind of effort, which not only is principled, but provides a large dataset that can fuel additional statistical analysis, computational modelling, and theoretical debate. The reviewers felt that emphasizing these issues even more in the manuscript would make clear how this kind of work contributes to the literature.

Each of the three reviewers provided extraordinarily long and detailed comments that were thoughtful and constructive. While we have tried to consolidate these comments, the result remains a quite long discussion, for which I apologize; however, given the care that went into these reviews, I decided it was helpful to preserve much of the content.

Essential revisions:

1) At the highest level, the theoretical importance of the original paper-and the current replication- needs to be even more clearly laid out. The interpretation of the original results is stated in only vague terms, which leads to confusion about what one should conclude. Crucially, the authors never really spell what mechanism would lead to effects on the N400 in response to an article preceding the putatively predicted noun (as opposed to the N1, P2, LAN, P600, or any other ERP component).

The current paper seems to argue for the "high-level compositional" (Introduction) view of the N400. The "N400 is elicited by every word of an unfolding sentence, and its amplitude is smaller (less negative) with increasing ease of semantic processing" (Introduction). Therefore, if a word does not fit semantically well within a context, N400 effects are expected. But DeLong et al., (2005) argued that the N400 reflects lexico-semantic prediction. The mixing of these two views is problematic here, as the DeLong et al., (2005) design "involved semantically identical articles (function words) rather than nouns or adjectives (content words) that are rich in meaning" (Introduction), which in turn would make "the observed N400 modulation by article-cloze.… unlikely to reflect difficulty interpreting the articles themselves" (Introduction). Unfortunately, this feature of the study makes it hard to make a precise prediction about what to expect at the level of processing the definite articles, especially if one is espousing a semantic integration view of the N400, for several reasons:

a) N400 effects are normally investigated in content words, not closed class words like articles.

b) As closed class words, articles elicit different ERP morphologies than open class words, including within the N400 time window.

c) Previous studies contrasting ERP responses to articles that are compatible or incompatible in upcoming word gender features elicited differences in late positivities, not N400.

d) The control study in the current manuscript shows that the incorrect pairing of a definite article allomorph with a subsequent word induces later positivities, not N400 effects. This calls into question why we should expect an N400 effect in the case of the predicted upcoming word.

e) There is a long history of research tying the N400 to general lexico-semantic processing, but not compositional semantics. At different places, the authors seem to argue both of these views. Under the compositional semantic integration view, the N400 may be affected by the phonological form of articles. Both forms contribute similarly to the semantics of the sentence (they are allomorphs after all). Semantically reversible sentences do not show N400 effects even when one order of the arguments is semantically odd (e.g.," the hunters chased the fox" vs. "the fox chased the hunters"), which argues against a compositional view of the N400. Given all this, why should one expect the N400 to be affected for allomorphic variations of the definite article when no obvious semantic integration effects are involved?

f) The fact that the a/an alternation is not triggered by the predicted head noun, but rather by the word immediately following the article, also confuses things quite a bit: why should there be, under any interpretation of the N400, an N400 effect for the alternating articles on the basis of the predicted head noun if the head noun is not necessarily (or even statistically) the trigger for the article form alternation? While the authors discuss this idea, it casts serious doubt on the original interpretation of the DeLong et al., findings.

These issues call into question the theoretical importance of the original study (and thus the replication). Contrary to what the authors state in the paper, that original paper offers only weak evidence for a "strong prediction view" precisely because the theoretical implications are vague and unspecified.

2) Throughout the paper, the authors emphasize the failure to find graded neural effects of cloze probability on the indefinite article preceding a noun. The reviewers do not believe this is as critical as it is played up in the manuscript. For instance, the current study also provides some evidence that a mismatch between articles (a/an) and subsequent nouns, conducted in the same participants tested in a subsequent experiment, leads to different neural activity. Moreover, a gradient in response at the level of group data (as in the original study) could nonetheless arise from discrete, all-or-none prediction at the single-trial level. It seems a bit disingenuous to argue that the study should call into question the importance of prediction on theories of language comprehension so broadly.

Relatedly, it would be good if results from studies of other "predictive" effects were considered and reconciled with the present findings. The authors list several of these in the Discussion section, but omit others; the authors should provide a balanced discussion of other evidence of different prediction effects, such as

- gender predictions (Wicha, Moreno and Kutas, 2004; Otten, Nieuwland and van Berkum, 2008; Van Berkum et al., 2005).

- predictions of initial sounds of upcoming words (Connolly and Phillips, 1994).

- expectations of the form of words with specific syntactic classes (Dikker et al., 2010).

- processes affected pre-N400 ERP responses driven by form-level characteristics (Lau et al., 2006).

- effects on early sensory responses like the M100 (Dikker et al., 2010; Dikker, Rabagliati, and Pylkkanen, 2009).

- behavioral evidence by close-shadowing (Marslen-Wilson, 1985).

The authors argue that the present null results for articles can "be straightforwardly reconciled with effects reported for other pre-nominal manipulations" (Discussion section), but studies like those above provide evidence for predictive processing. The preceding paragraph (Discussion section) offer a number of caveats, but readers of this paragraph might take away the message that the current results "do not necessarily exclude phonological form pre-activation" rather than the message from previous page (Discussion section) that "there is currently no clear evidence to support routine probabilistic pre-activation of a noun's phonological form during sentence comprehension.

In general, the wording of the manuscript should be revised throughout so as to not overstate the implications of the findings, while at the same time giving a consistent message about what the findings do mean.

3) The correlation analysis in the original paper analyzed only 10 data points, each of which is a mean across 32 participants who each did ~80 trials (binned into deciles based on cloze probability). This analysis excludes several critical sources of variance between trials, items, and participants, which increases the likelihood of type I error. On the other hand, the analysis is also underpowered, since with only 10 data points and 9 degrees of freedom, statistical significance requires an exceptionally strong correlation; thus, the likelihood of type II error is great (making it not very surprising that results fail to replicate). This is an important lesson for the field and should be brought our more clearly in the present manuscript. While adding additional analysis is not necessary for the manuscript to be publishable, this point could be strengthened, for instance, by running simulations to assess the power of the original DUK study, varying the number of participants and items and exploring whether a graded effect of cloze probability on article and noun processing can be detected.

4) On a related note, the DUK study also reports a reliable difference in N400 magnitude between high and low cloze probability articles and nouns (Figure 1A of DUK, left panel); their analysis included by-participant variation as a random effect. Given their counterbalanced design and arguments from Raaijmakers et al., (1999), this analysis is less prone to errors than their correlational analysis. The categorical analysis provides additional evidence for a role of prediction – albeit not a graded one. Figure 3 of the present manuscript presents similar categorical results, but there is no statistical analysis reported. Do these new results also show significant evidence for a non-graded prediction of articles? If so, this would substantially modify the conclusions drawn from the current study. This should be considered and discussed.

5. The reviewers noted that there is an online critique and re-analysis posted on BioRxiv by Yan, Kuperberg and Jaeger (https://www.biorxiv.org/content/early/2017/05/30/143750) that endorses the importance of the issues raised and the value of the data collected here. It's not clear whether and how the authors anticipated this commentary in the present manuscript.

For instance, this commentary suggests that Bayesian surprisal, KL divergence between pre- and post-article word probability distributions, or other variables might show a closure correspondence to the neural data than does cloze probability; both the current authors and the original study make an implicit assumption that cloze probability is the best predictor.

Another example is the idea raised in the commentary that there is a chain of cognitive processes mediating between meaning predictions and form-based predictions for indefinite articles. A failure, probabilistic limitation, or probability mis-estimation (e.g., cloze tests overestimating the probability of an indefinite article) in these mediating processes may explain why the strongly constraining lexical/semantic predictions set-up by the sentence contexts don't always influence the processing of indefinite articles. The sentence contexts set up semantic expectations, which constrain the choice of lemmas in critical sentence positions. Some predicted lemmas have phonological forms that have specific consequences for articles; however, form-based predictions may not follow even if a semantically predicted lemma is chosen (e.g., "a kite" vs "an aeroplane", but also "an old kite" or "a plane"). The authors acknowledge this point (Discussion section), but do not carry this line of argument through the rest of the paper. It's not the case that a failure to observe predictions at the level of the phonological form of indefinite articles necessarily implies that all prediction is absent; yet in a number of places they seem to argue along these lines.

The manuscript's impact would be heightened by incorporating comments that address points like these that were raised in this online critique.

6) It is great that the authors included Bayes factor analyses to gain theoretical insights from null results. These analyses provide evidence in favor of the null hypothesis that there are no graded predictions for articles, despite numerical effects in the expected direction. However, these analyses depend on assumptions about the expected effect size and other details that are not clearly spelled out. These priors should be provided. For instance, Figure 1A/B DUK suggests that the expected effect size of the cloze probability effect on the article is considerably smaller than the effect on the noun itself. It seems that this was taken into account in Bayes factor computations (given the graphs in SI Figure 2), but what other assumptions are made in this analysis? Was the variance expected to be equivalent for articles and nouns? Can the authors justify the choice of using a Gaussian? What does the estimated effect size being greater than zero mean for the "pre article" period (in SI Figure 2)? If predictive processing occurs, then predictions should have been computed prior to the onset of the article.

7) The use of a pre-article baseline correction may be problematic, since neural responses prior to the onset of the article will include processing of linguistic information that contributes to the generation of predictions or (equivalently) constrains the likely form and meaning of upcoming words. For example, recent data from Grisoni et al., (2017) suggests a "semantic readiness potential" reflecting pre-activation of specific semantic representations. Similar semantic pre-activation is likely present in the sentences used in the current study. By doing a pre-article baseline, such differences could corrupt or conflate with differences ascribed to neural responses to the article. The authors consider but do not describe or discuss these effects in Supplementary figure 2. While the authors are not responsible for the fact that the original study used an unconventional baseline, in subsection “Single-trial analysis” the manuscript implies that there is a reason why it is appropriate, because the current authors observed non-significant cloze effects immediately prior to the critical noun.

At a minimum, this potential confound should be raised and discussed. Better still, the authors could run an analysis in which they either use a pre-sentence baseline or analyze raw ERPs without baseline correction and compare to the pre-article baseline analysis. They could also analyze whether there are differences to highly vs weakly constraining sentences prior to the onset of the article (building on the analysis in SI Figure 2), an approach that might benefit from using a different predictor variable the captures the strength of prediction (e.g. the entropy of the cloze probability distribution) rather than not whether or not the prediction is subsequently confirmed or violated. Inclusion of these types of additional analyses could support a more nuanced discussion of whether or not participants compute the likely meaning of upcoming words.

8) The distinction between form and meaning predictions may influence differences or similarities between N400 effects seen on articles and on nouns. The original DUK paper reports similar magnitude and topography of N400 effects on both the article and noun. It would be instructive to compare the location, timing, and effect size for cloze probability correlations between the article and the noun in the current, larger dataset. Additional tests for difference between the significant and null correlations would be valuable. For instance, is the reliable effect of graded cloze probability on nouns significantly different from the null effect of cloze probability on articles? This difference should be reliable if semantic integration is the key driver of N400 effects on nouns.

In a similar vein, what is the assumed model of the N400 generator? There are at least two published computational models that seek to explain the large body of N400 effects in the existing ERP literature. One proposes something akin to integration demands (Cheyette and Plaut, 2016), another proposes a form of (semantic) prediction error (Rabovsky and McRae, 2014). Such models provide a helpful context in which to discuss whether or not prediction violations at the level of form will lead to an N400 in the absence of any ongoing semantic integration demands. Ultimately, questions about the replicability or otherwise of the data presented by DUK are only informative to the extent that they are or, are not theoretically constraining. A discussion of these and other computational theories of the N400 are important for interpreting the present results, and how these results should be considered in the field of language comprehension going forward.

Other general comments for the authors to consider:

1) The authors conducted a single-trial analysis using linear mixed effects regression (LMER) analysis. This is the most appropriate analysis method since it includes both between-participant and between-item variance and combines all of the usable data from individual participants and single trials into one analysis. While LMER analysis is now preferred to correlation analysis used in the original study, it only came to prominence after the publication of the original paper. Pointing out the advantages of LMER analyses would enhance the current paper's impact. For instance, especially when analyzing effects of continuous predictor variables, LMER may offer better control over false-positives than conventional analyses. This is an important message for researchers studying the neuroscience of language, who generally do not undertake this approach.

2) The flip side of using single-trial LMER, however, is that it is not clear whether or not the link between cloze probability and neural responses is linear or logistic. The independent measure (cloze probability) used as a predictor variable is summarizing a binary outcome variable (i.e., did individual participants in the cloze test generate the article "a" or "an") as a probability. It is at least plausible that neural outcomes could be similarly binary: participants predict "a" or "an" and then don (or don't) generate a neural prediction error when an unexpected word is presented. Would reliable effects of cloze probability be observed with a logistic or binomial linking function? While this is not the form of graded prediction tested by DUK, it seems quite likely that averaging over many participants and many trials (as in the DUK correlation analysis) would generate a graded, linear relationship between cloze probability and neural outcomes, even if the underlying relationship was logistic or non-linear. This possibility should be considered, assessed and discussed.

3) Previous work has explored the presence/absence of fillers, which is another difference between the DUK study and the current one. While it is unlikely to explain the failure to replicate results, some additional discussion seems appropriate. Given the large scale of the present dataset there should be many opportunities for determining whether cloze probability effects (for nouns, if not for articles) are enhanced once participants have already encountered a number of distinctive, high/low cloze probability sentences.

4) The fact that the nine labs inadvertently manipulated the timing of stimulus presentations of the critical articles and nouns (see footnote 3, Materials and methods section) is unlikely critical to their findings/conclusions. However, it would be useful to know whether the timing of early, visual form responses for written words that are delayed and non-delayed differ. Simply arguing that the change in timing is "unlikely to be noticed" and "after the N400 window" is making assumptions that can and should be tested with their ERP data.

5) The use of different baselines was confusing to multiple reviewers. A table that summarizes details of all the current analyses and how these correspond to the procedures used in the original DUK study would help the reader keep straight the different results.

6) Both the original and current studies present a high percentage of a/an continuations in the cloze test. Could this procedure bias participants towards using a/an rather than "his", "hers", "their," etc. that aren't marked for the phonological form of the subsequent noun? After encountering several sentences in which the indefinite article is appropriate, participants may routinely complete the remaining sentences with an indefinite article continuation due to this sort of syntactic priming. A comment about this would be welcome.

7) In multiple places, the authors use phrasing about how the study argues for "a more limited role for prediction during language comprehension." This wording seems overly broad, conflating very different conceptions of prediction across different levels of representation. There is a deep theoretical division between proposed mechanisms of conceptual pre-activation (e.g., the concept of 'readiness' in the discourse literature, linked to the N400 literature beautifully in Jos Van Berkum's work) and mechanisms of predicting the form of the upcoming input (commitment to a particular linguistic formulation of the message). It seems likely that the former, long argued for in the discourse literature, might play a central role in language comprehension, while contributions from the prediction-of-form mechanism may be quite limited. It would be unfortunate if readers take the current study to cast doubt on both extremely different mechanisms, when it actually only speaks to the latter mechanism. Please consider rewording this throughout.

---

## [Author Response]

Essential revisions:1) […] These issues call into question the theoretical importance of the original study (and thus the replication). Contrary to what the authors state in the paper, that original paper offers only weak evidence for a "strong prediction view" precisely because the theoretical implications are vague and unspecified.

We thank the reviewer for these comments. In the Introduction we now state “DeLong et al., presented the systematic, graded N400 modulation by article-cloze as strong evidence that participants activated the nouns and articles in advance of their appearance, and that the disconfirmation of this prediction by the less-expected articles resulted in processing difficulty (higher N400 amplitude at the article)”

This reviewer questions the theoretical relevance offered by DeLong et al., about their own findings and points out different interpretations of the N400 that may have different repercussions for interpreting the DUK results. However, that interpretation exercise assumes that the original effects were reliable and replicable. The motivation of our study was to test whether the effects can be replicated in the first place, after which a theoretical interpretation becomes relevant, as we offer in our discussion. This reviewer suggests that we are advocating a particular interpretation of the N400. This is not the case. Our group does not take the N400 as a pure measure of high-level semantic composition. We purposefully used the neutral term “ease of semantic processing”, which can include both memory access and integration processes. Our only assumption regarding the N400 followed the logic of DeLong et al., namely that noun-elicited N400s themselves do not yield unambiguous evidence for prediction.

We certainly agree that DUK offers only weak evidence for the “strong prediction view”, in particular because their effects are weak, and they tried multiple analyses. We now refer to the original results throughout the paper as “the most acclaimed” instead of “the strongest evidence”. This is because while DUK may offer only weak evidence, that is not how their results have been and are still received in the wider literature: indeed, several impactful review papers literally cite DUK as ‘strong evidence’. This reviewer may call into question the theoretical relevance of the original study, but, as we describe in our text, the original study has had a major impact on the field, is cited in all the major theoretical review papers on linguistic prediction. Moreover, never has a question been raised about the replicability or interpretation of the DUK findings (with a few exceptions which we cite in the text).

This reviewer also states that all pre-nominal gender manipulations elicited effects other than N400 effects. This is not the case. Prenominal gender manipulations have elicited a range of effects, including N400 effects (e.g. Otten and Berkum, 2009; Wicha et al., 2003; Kochari and Flecken, under review) and sometimes P600-like effects or frontal negative ERP effects. We mentioned this in the previous submission and have now expanded this section. However, why different effects are observed for those gender-manipulations is unknown and beyond the scope of our paper on a different manipulation.

2) Throughout the paper, the authors emphasize the failure to find graded neural effects of cloze probability on the indefinite article preceding a noun. The reviewers do not believe this is as critical as it is played up in the manuscript. For instance, the current study also provides some evidence that a mismatch between articles (a/an) and subsequent nouns, conducted in the same participants tested in a subsequent experiment, leads to different neural activity. Moreover, a gradient in response at the level of group data (as in the original study) could nonetheless arise from discrete, all-or-none prediction at the single-trial level. It seems a bit disingenuous to argue that the study should call into question the importance of prediction on theories of language comprehension so broadly.

We added comments to clarify that we do not question the general importance of prediction in theories of language (see also our response to the next parts of comment 2). Here and in M4, the reviewers bring up the use of cloze as a dichotomous variable. We have performed this analysis with a median-split approach (100 and 500 ms baseline for the articles, 100 ms baseline for the nouns). The new code is also available on our OSF page. Importantly, the results are not affected substantially. Moreover, these analyses all yielded lower chi-sq values (and higher p-values) than the continuous approach. We mention these analyses and their results in the Results section.

Relatedly, it would be good if results from studies of other "predictive" effects were considered and reconciled with the present findings. The authors list several of these in the Discussion section, but omit others; the authors should provide a balanced discussion of other evidence of different prediction effects, such as- gender predictions (Wicha, Moreno and Kutas, 2004; Otten, Nieuwland and van Berkum, 2008; Van Berkum, et al., 2005).- predictions of initial sounds of upcoming words (Connolly and Phillips, 1994).- expectations of the form of words with specific syntactic classes (Dikker, et al., 2010).- processes affected pre-N400 ERP responses driven by form-level characteristics (Lau et al., 2006).- effects on early sensory responses like the M100 (Dikker et al., 2010; Dikker, Rabagliati, and Pylkkanen, 2009).- behavioral evidence by close-shadowing (Marslen-Wilson, 1985).

We thank the reviewer for the suggestion. We already cite many of the relevant papers on gender manipulations, which are the most relevant because they involve prenominal manipulations like DUK. We mentioned that these studies have not elicited a consistent pattern of results and have sometimes demonstrated non-replicability. However, providing a review of all ERP effects associated with prediction is beyond the scope of our work, and there are already many other papers who review this literature. This reviewer appears to take all the listed studies as clear-cut evidence for some form of prediction. But many of these studies are problematic for all sorts of reasons, including the issue of replicability and data-contingent analysis. For example, the Connolly and Phillips results are intriguing but rely entirely on the distinction between the N200 and N400, which is poorly defined in the original study, and not observed in many other studies (Van Petten et al., 1999; Van den Brink et al.,; Boudewyn et al., 2015; Diaz and Swaab, 2007). The study by Lau et al., reported ELAN-results that were not necessarily pre-N400 (200-400 ms), could only be obtained with a common average procedure that no other ELAN study has ever used, showed clear evidence of a ‘polar average reference effect’, and has failed to replicate (Kaan, Kirkham and Wijnen, 2016). The Dikker et al., studies are low N (~12) studies and p-values in the.01-.05 range with many researcher degrees of freedom due to the methodology, and do not show any M1 modulation by predictability. We also note that all the studies mentioned by the reviewer are single studies that, like DUK, have not been directly replicated, hence the robustness of the reported findings needs to be verified, as well as their generalizability to other contexts, labs, etc. In other words, all these observations are worthy of a dedicated, thorough and critical review, but that is for a different paper. Our aim is not to question the general role of prediction in language comprehension (see next point), but to illustrate the importance of replication.

The authors argue that the present null results for articles can "be straightforwardly reconciled with effects reported for other pre-nominal manipulations" (Discussion section), but studies like those above provide evidence for predictive processing. The preceding paragraph (Discussion section) offer a number of caveats, but readers of this paragraph might take away the message that the current results "do not necessarily exclude phonological form pre-activation" rather than the message from previous page (Discussion section) that "there is currently no clear evidence to support routine probabilistic pre-activation of a noun's phonological form during sentence comprehensionIn general, the wording of the manuscript should be revised throughout so as to not overstate the implications of the findings, while at the same time giving a consistent message about what the findings do mean.

We do not question the existence of predictive processing in general, and we now make that more explicit in the revised Discussion section(“Our results, however, should not be taken as evidence against prediction in language processing more generally, and we believe that prediction could play an important role in language comprehension”). We also note that each of these studies argues for a slightly different aspect of prediction, and none of them clearly supports "routine probabilistic pre-activation of a noun's phonological form during written sentence comprehension". Furthermore, as pointed out above, these studies are not as strong as they may first appear, for all sorts of reasons. As for our own results, they are not necessarily evidence against phonological form pre-activation, just not very clear evidence for it; that is what we would like readers to take away from our discussion.

3) The correlation analysis in the original paper analyzed only 10 data points, each of which is a mean across 32 participants who each did ~80 trials (binned into deciles based on cloze probability). This analysis excludes several critical sources of variance between trials, items, and participants, which increases the likelihood of type I error. On the other hand, the analysis is also underpowered, since with only 10 data points and 9 degrees of freedom, statistical significance requires an exceptionally strong correlation; thus, the likelihood of type II error is great (making it not very surprising that results fail to replicate). This is an important lesson for the field and should be brought our more clearly in the present manuscript. While adding additional analysis is not necessary for the manuscript to be publishable, this point could be strengthened, for instance, by running simulations to assess the power of the original DUK study, varying the number of participants and items and exploring whether a graded effect of cloze probability on article and noun processing can be detected.

We appreciate the suggestion to emphasize the potential problems with the correlation approach. The problems with this approach are clear. In our study, for example, while a statistically significant effect at the noun was found in each participating lab using a more powerful single-trial analysis, the effect across labs in the correlation analysis was mixed. We have now added a statement to that effect in the Discussion section (“Moreover, our single-trial analysis revealed a significant noun-cloze effect in each of the laboratories, further demonstrating that our single-trial analysis is a more powerful approach than the averaged-based correlation approach of DeLong et al.”). Given the obvious problems with the correlation-analysis, we feel that focusing too much on that analysis would detract from our own single-trial results. In addition, we feel that running simulations to assess the power of DUK with varying participant and item numbers is unnecessarily burdensome, if all that is available to us is the correlation coefficients: running such simulations using the actual DUK data would be a lot more meaningful, but they have refused to share their data so far. In addition, we know that DUK have analyzed their own data with a single-trial analysis, which is a more informative analysis given the weaknesses of the original analysis, and which also did not yield a statistically significant article-effect. We assume that DeLong et al., will publish those new results in the near future, so that power-analyses can be applied to the improved analyses.

4) On a related note, the DUK study also reports a reliable difference in N400 magnitude between high and low cloze probability articles and nouns (Figure 1A of DUK, left panel); their analysis included by-participant variation as a random effect. Given their counterbalanced design and arguments from Raaijmakers et al., (1999), this analysis is less prone to errors than their correlational analysis. The categorical analysis provides additional evidence for a role of prediction – albeit not a graded one. Figure 3 of the present manuscript presents similar categorical results, but there is no statistical analysis reported. Do these new results also show significant evidence for a non-graded prediction of articles? If so, this would substantially modify the conclusions drawn from the current study. This should be considered and discussed.

DUK figure 1a showed the difference between high and low only as “illustrative ERPs”, the original report did NOT report any categorical statistical analysis. However, DeLong et al., (2009) DID report such an analysis and the effect of cloze on article-elicited ERPs was not significant. We have also performed these analyses, as described in M2.

5) The reviewers noted that there is an online critique and re-analysis posted on BioRxiv by Yan, Kuperberg and Jaeger (https://www.biorxiv.org/content/early/2017/05/30/143750) that endorses the importance of the issues raised and the value of the data collected here. It's not clear whether and how the authors anticipated this commentary in the present manuscript.For instance, this commentary suggests that Bayesian surprisal, KL divergence between pre- and post-article word probability distributions, or other variables might show a closure correspondence to the neural data than does cloze probability; both the current authors and the original study make an implicit assumption that cloze probability is the best predictor.

We are aware of this unpublished critique and re-analysis by YKJ, which is a comment on a previous draft of our manuscript that became available on Biorxiv in February 2017. We did not incorporate discussion of this work because we felt that a separate, dedicated response was more appropriate, in which we could extensively comment on some of the details of their analysis and their argumentation (after all, their commentary is 60 pages long). YKJ bring up an interesting point, namely that the cloze variable should be log-transformed, based on their account in terms of Bayesian surprisal, which itself is not a new argument (e.g. Smith and Levy, 2013). We have now added a brief discussion of the Yan et al. commentary. To retain a balance between succinctness and completeness in the main text, we moved our fuller response to an online supplement on our OSF page with a link in the text (https://osf.io/mb2ud/).

In essence, the YKJ conclusion is not different from what we conclude, namely that there is some evidence for a non-zero effect. However, YKJ ignore our strong evidence that the effect is not like that of DUK, and misrepresent our conclusions to argue their case (e.g., “their conclusion that their data provides no evidence for prediction on the article”, while we in fact purposefully refrained from using terms like “no evidence”, we used terms like “no convincing evidence” or “no clear evidence”). The YKJ argument about how cloze probabilities should be analyzed is ultimately one for further testing, because their exploratory analysis was conditional on our data. YKJ applaud us for “leading by example in pre-registering their study” but, have yet to follow our example. It is one year since we published our pre-print, and about 10 months after Yan et al., performed their analyses. As far as we are aware, they have not proceeded to test their hypothesis in a pre-registered study, nor have they attempted to reanalyze the wealth of N400 data from their own labs. YKJ report no independent data to support the conclusion that Bayesian surprisal is a better predictor of N400 activity than cloze probability. Unlike YKJ, we have run their transformed analysis on other cloze datasets. The results speak against a generally better fit of log-cloze, and we have also made those data available online.

The YKJ analysis did yield a significant article-cloze effect in our data, and a model with log-transformed cloze had a lower AIC (an indication of model fit) than a model with regular cloze. For the nouns, the transformation yielded a slightly higher T value for cloze. According to one of the current reviewers (S11): “No valid conclusion can be drawn from a change in numerical magnitude or a change in the observed p-value”, and if that is true the same concern should apply to YKJ as well. The article-effect becomes significant whereas it was not-significant before, but the difference between significant and not-significant itself may not be significant (Gelman and Stern, 2006). Moreover, their analysis could be problematic because between-item differences in the lower cloze scale are not matched on relevant variables. This is also suggested by the fact that their analysis also amplifies the effect in the pre-article time window (a marginally significant effect at p=0.058), so it might just be ‘amplifying noise’ (by amplifying the between-item differences at the lower end of the cloze scale), and does not yield a significant effect when we used the original DUK baseline. Their results also hinge entirely on the zero-cloze values which they argue cannot be reliably estimated. Jaeger mentioned in a private message that “all the signal is in the zero-cloze values”, and this is correct; in fact, if one repeats their analysis without zero-cloze values the effect size changes direction (higher cloze leads to more negative voltage). In other words, the transformation seems to matter very little, and what it does seem to do is boost the impact of zero-cloze values which are problematic because they are indeterminate, so the evidence for their case is not very convincing.

YKJ present the significant effect of article-cloze in their main text, and they put the analyses that do not yield as clear support in a footnote, and do not report analysis of the pre-article window. This seems somewhat disingenuous to us, given that we reported all analyses in the main text, significant or non-significant. YKJ also report their analysis as “the only one we performed”, although there is no record of this and cannot be verified. In fact, private twitter messages by Jaeger to Nieuwland stated the opposite: that multiple analyses (to deal with the zero-cloze smoothing issue) were tried, and although they were said to lead to similar results, those are not mentioned in YKJ, and the reported ‘correction for multiple comparisons’ can therefore not be complete. A similar concern can be raised against the other analyses that were tried: for instance, YKJ analyzed the data with and without lab as random effect and they analyzed the data to test effects in each individual lab (with and without log transformation, so 2*9=18 comparisons). In none of those analyses did they find a statistically significant effect, despite what they state (“we find a significant effect on both the noun and the article in 8 out of 9 of the labs that participated in the replication attempt”). In addition, none of these analyses were taken into account during the correction for multiple comparisons. All this merely demonstrates the need for YKJ to specify and pre-register their analysis in advance and run a confirmatory study to test their hypothesis.

Finally, what strikes us as odd is that YKJ have many things to say about why our analysis is sub-optimal and our conclusions should not be accepted. But when it comes to the DUK data, all they say is “For the purpose of this discussion, we take the effect that they report at face value”, despite the many problems with the DUK analysis that we had identified. The importance of the log-transform as suggested by YKJ would also need to apply to the Delong study, and it's not clear that the transform there would strengthen their effect rather than weaken it.

Another example is the idea raised in the commentary that there is a chain of cognitive processes mediating between meaning predictions and form-based predictions for indefinite articles. A failure, probabilistic limitation, or probability mis-estimation (e.g., cloze tests overestimating the probability of an indefinite article) in these mediating processes may explain why the strongly constraining lexical/semantic predictions set-up by the sentence contexts don't always influence the processing of indefinite articles. The sentence contexts set up semantic expectations, which constrain the choice of lemmas in critical sentence positions. Some predicted lemmas have phonological forms that have specific consequences for articles; however, form-based predictions may not follow even if a semantically predicted lemma is chosen (e.g., "a kite" vs "an aeroplane", but also "an old kite" or "a plane"). The authors acknowledge this point (Discussion section), but do not carry this line of argument through the rest of the paper. It's not the case that a failure to observe predictions at the level of the phonological form of indefinite articles necessarily implies that all prediction is absent; yet in a number of places they seem to argue along these lines.The manuscript's impact would be heightened by incorporating comments that address points like these that were raised in this online critique.

We have added an explicit statement that our results are not to be taken as evidence against prediction in general (see response to Essential revision #2); we never said that our results suggest ‘prediction is absent’ and it was not our intention to imply this. As this reviewer writes, our submission already mentioned that the article-form does not rule out the expected meaning altogether. This aspect of the a/an rule in part triggered our interest in the DUK study, which we take up in the Discussion section. We have added a mention of how the a/an rule works on page 4 but we do not see why this would have to be brought up “throughout the rest of the paper”. We wrote it as a potential explanation for the discrepant findings, and this was available in our initial pre-print (available online in Feb 2017), and was also discussed earlier in Ito, Martin and Nieuwland (2017; available online in May 2016), so all in advance of the Yan et al. commentary.

6) It is great that the authors included Bayes factor analyses to gain theoretical insights from null results. These analyses provide evidence in favor of the null hypothesis that there are no graded predictions for articles, despite numerical effects in the expected direction. However, these analyses depend on assumptions about the expected effect size and other details that are not clearly spelled out. These priors should be provided. For instance, Figure 1A/B DUK suggests that the expected effect size of the cloze probability effect on the article is considerably smaller than the effect on the noun itself. It seems that this was taken into account in Bayes Factor computations (given the graphs in SI Figure 2), but what other assumptions are made in this analysis? Was the variance expected to be equivalent for articles and nouns? Can the authors justify the choice of using a Gaussian?

This reviewer may have missed our description of the priors in the methods section. Yes, they differed for articles and nouns because they were based on the DeLong correlation results. We have seen the LMEM results of the DUK data, but DeLong et al. refused to share their LMEM based priors for our analysis. We can inform the reviewer that those priors pulled our estimates closer to zero. Yes, variance was assumed to be equivalent. There is no ground to suspect a Gaussian would not be appropriate. Of note, these are all exploratory, non-preregistered analyses and the code and data are online so that anyone can explore our data further.

What does the estimated effect size being greater than zero mean for the "pre article" period (in SI Figure 2)? If predictive processing occurs, then predictions should have been computed prior to the onset of the article.

In short, we don’t really know. We analyzed this window because we found out, after publishing our pre-print, that the 500 ms window was used for baseline correction by DeLong. Their choice differed from most of the other studies in their lab, so maybe they picked this window to correct for pre-article differences, we do not know. We also saw what looked like slow drift effects in the ERPs in some labs in this window. The effects in this window cannot be meaningfully interpreted. This is because the two versions of each item are identical before the article. So, any effect there must be a mix of noise and effects due to differences between the sentence contexts (e.g., the predictability or integrability of the pre-article word, sentence position, lexical effects; all of these were not matched between the sentence contexts). It is possible that the distributions of the unexpected and expected articles along the cloze scale is somehow interacting with some of these between-item sentence context differences. We agree that predictions should be generated before the article, but this particular design does not allow for a meaningful analysis.

7) The use of a pre-article baseline correction may be problematic, since neural responses prior to the onset of the article will include processing of linguistic information that contributes to the generation of predictions or (equivalently) constrains the likely form and meaning of upcoming words. For example, recent data from Grisoni et al., (2017) suggests a "semantic readiness potential" reflecting pre-activation of specific semantic representations. Similar semantic pre-activation is likely present in the sentences used in the current study. By doing a pre-article baseline, such differences could corrupt or conflate with differences ascribed to neural responses to the article. The authors consider but do not describe or discuss these effects in Supplementary figure 2. While the authors are not responsible for the fact that the original study used an unconventional baseline, in subsection “Single-trial analysis” the manuscript implies that there is a reason why it is appropriate, because the current authors observed non-significant cloze effects immediately prior to the critical noun.At a minimum, this potential confound should be raised and discussed. Better still, the authors could run an analysis in which they either use a pre-sentence baseline or analyze raw ERPs without baseline correction and compare to the pre-article baseline analysis. They could also analyze whether there are differences to highly vs weakly constraining sentences prior to the onset of the article (building on the analysis in SI Figure 2), an approach that might benefit from using a different predictor variable the captures the strength of prediction (e.g. the entropy of the cloze probability distribution) rather than not whether or not the prediction is subsequently confirmed or violated. Inclusion of these types of additional analyses could support a more nuanced discussion of whether or not participants compute the likely meaning of upcoming words.

We appreciate this concern about using a baseline window period in which predictive processing takes place. The reviewer correctly points out an inherent problem with choosing an appropriate baseline for the article in a study that searches for the presence of prediction for the article. We performed the requested analysis without baseline correction, this does not change the observed pattern (below) and introduced a large amount of variance into the data that a baseline procedure is supposed to remove.

Fixed effects:

Estimate Std. Error df t value Pr(>|t|)cloze-effect 0.1351 0.3897 474.5000 0.347 0.729

We emphasize that we are not arguing against some form of semantic prediction, and our revised draft makes this more clear. While this reviewer wants additional exploratory analyses to support a discussion of whether or not word meaning was anticipated, that was never the question of our study, and there is already quite a lot of evidence that people predict the meaning of upcoming words. Furthermore, the 80 sentence contexts are not matched on relevant variables known to impact EEG activity, so the experimental design is not suitable for addressing the question of pre-article constraint. We considered it important to make this clearer to the readers. In subsection “Exploratory (i.e., not pre-registered) single-trial analyses” we added “Because the sentence context of each item was identical for the expected and unexpected article, effects in the pre-article window cannot be meaningfully related to the appearance of the article. Effects in this window must therefore be due to a spurious mix of ‘residual EEG background noise’ (activity that differed between expected and unexpected conditions but was unrelated to actual expectancy) with EEG activity associated with the specific word appearing before the article (which varied between items in terms of lexical characteristics, contextual constraint, and sentence position).”

8) The distinction between form and meaning predictions may influence differences or similarities between N400 effects seen on articles and on nouns. The original DUK paper reports similar magnitude and topography of N400 effects on both the article and noun. It would be instructive to compare the location, timing, and effect size for cloze probability correlations between the article and the noun in the current, larger dataset. Additional tests for difference between the significant and null correlations would be valuable. For instance, is the reliable effect of graded cloze probability on nouns significantly different from the null effect of cloze probability on articles? This difference should be reliable if semantic integration is the key driver of N400 effects on nouns.

We already compare the effects of articles and nouns in our more powerful single trial-analysis, it shows a massive difference. We have also seen a single-trial re-analysis of the DUK data (by DUK), it shows the same pattern. We do not see the added value of the suggested analysis. We don’t think it is particularly instructive to know whether the correlations differ if this analysis is clearly problematic. Moreover, the noun-correlations are significantly different from zero, whereas the article-correlations go into the opposite direction at the channels where the noun-N400 effects are strong.

In a similar vein, what is the assumed model of the N400 generator? There are at least two published computational models that seek to explain the large body of N400 effects in the existing ERP literature. One proposes something akin to integration demands (Cheyette and Plaut, 2016), another proposes a form of (semantic) prediction error (Rabovsky and McRae, 2014). Such models provide a helpful context in which to discuss whether or not prediction violations at the level of form will lead to an N400 in the absence of any ongoing semantic integration demands. Ultimately, questions about the replicability or otherwise of the data presented by DUK are only informative to the extent that they are or, are not theoretically constraining. A discussion of these and other computational theories of the N400 are important for interpreting the present results, and how these results should be considered in the field of language comprehension going forward.

The two cited models are not models of sentence-level comprehension effects in N400 activity but of word-level comprehension, so it's difficult to say what they would predict on DUK type violations (if anything). This reviewer states that “these and other computational theories of the N400 are important for interpreting the present results”, but none of these models has ever said anything about ERP effects of pre-nominal manipulations in prediction-studies like DUK, and these studies have nothing or very little so say about form-prediction (Rabovsky et al., 2018), probably because they all involve purely semantic representations. It is not at all clear to us how these theories help us understanding whether the true population-level effect size is more like that reported by DUK or more like that reported here.

Cheyette and Plaut assume that the N400 ERP component reflects difficulty of semantic access, essentially the view-point of Kutas et al. Rabovsky and McRae argue that the N400 reflects semantic surprise, and a later model about sentence-level comprehension by Rabovsky, Hanssen and McClelland “treats N400 amplitudes as indexing the change induced by an incoming word in an implicit probabilistic representation of meaning”, and does not assume separate stages of access and integration. All these accounts assume the N400 component reflects a unitary process, which we think is unlikely, and, in fact, our own new results suggest that EEG activity in the N400 window is sensitive to different types of information at different times (Nieuwland et al., 2018). The only computational model that is accompanied by discussion of DUK-like effects is from Fitz and Chang (https://psyarxiv.com/frx2w/) which takes the N400 as a signal of error propagation. Fitz and Chang specifically discuss the fact that their model does not capture DUK-like N400 effects as a limitation of their model, and state that DUK-like effects would require a different set of mechanisms beyond their own model.

Other general comments for the authors to consider:1) The authors conducted a single-trial analysis using linear mixed effects regression (LMER) analysis. This is the most appropriate analysis method since it includes both between-participant and between-item variance and combines all of the usable data from individual participants and single trials into one analysis. While LMER analysis is now preferred to correlation analysis used in the original study, it only came to prominence after the publication of the original paper. Pointing out the advantages of LMER analyses would enhance the current paper's impact. For instance, especially when analyzing effects of continuous predictor variables, LMER may offer better control over false-positives than conventional analyses. This is an important message for researchers studying the neuroscience of language, who generally do not undertake this approach.

We agree. In the Materials and methods section on our single-trial analysis, we added “Especially when analyzing effects of a continuous predictor variable such as cloze probability, LMER offers better control over false-positive results than the averaged-based correlation analysis of the original.” In the last paragraph of the Introduction, we added “and a single-trial analysis that modelled variance at the level of item and subject (with a linear mixed-effects model, which offers better control over false-positives than the replication analysis when analyzing effects of the continuous predictor cloze probability.”

2) The flip side of using single-trial LMER, however, is that it is not clear whether or not the link between cloze probability and neural responses is linear or logistic. The independent measure (cloze probability) used as a predictor variable is summarizing a binary outcome variable (i.e., did individual participants in the cloze test generate the article "a" or "an") as a probability. It is at least plausible that neural outcomes could be similarly binary: participants predict "a" or "an" and then don (or don't) generate a neural prediction error when an unexpected word is presented. Would reliable effects of cloze probability be observed with a logistic or binomial linking function? While this is not the form of graded prediction tested by DUK, it seems quite likely that averaging over many participants and many trials (as in the DUK correlation analysis) would generate a graded, linear relationship between cloze probability and neural outcomes, even if the underlying relationship was logistic or non-linear. This possibility should be considered, assessed and discussed.

To our knowledge, these analyses would require a binomial dependent variable, it is unclear how this would be applied to our data. But, regardless, this could be an interesting alternative analysis that others could use to explore our data (and later confirm its results in subsequent testing). We have discussed the issue of alternative analyses in our revised Discussion section.

3) Previous work has explored the presence/absence of fillers, which is another difference between the DUK study and the current one. While it is unlikely to explain the failure to replicate results, some additional discussion seems appropriate. Given the large scale of the present dataset there should be many opportunities for determining whether cloze probability effects (for nouns, if not for articles) are enhanced once participants have already encountered a number of distinctive, high/low cloze probability sentences.

We appreciate the interest in this issue of fillers. It is unclear what additional discussion this reviewer would like to see. We know that DeLong et al., also have run other experiments with the a/an manipulations with different kinds of fillers. Publication of those results will offer a better opportunity to determine the importance of fillers. In our own dataset, there are opportunities to examine effects of experiment-position, and, in fact, when we posted our BioRxiv pre-print about a year ago, Florian Jaeger already notified us of his intentions to perform such an analysis using the data we made available. Our data is available to anyone with an interest in pursuing such an exploratory analysis.

4) The fact that the nine labs inadvertently manipulated the timing of stimulus presentations of the critical articles and nouns (see footnote 3, Materials and methods section) is unlikely critical to their findings/conclusions. However, it would be useful to know whether the timing of early, visual form responses for written words that are delayed and non-delayed differ. Simply arguing that the change in timing is "unlikely to be noticed" and "after the N400 window" is making assumptions that can and should be tested with their ERP data.

We have rephrased the description in this footnote to make it more clear. First, as the original footnote stated, the timing was slightly off only in 4 labs, not all labs, as this reviewer writes. We have also analyzed only the data where the timing was as intended, which did not change the results, and we had mentioned this already in the footnote. Second, the articles were never themselves delayed, but the blank screen that followed them was 80 ms longer than intended. This is crucial, because it means that the first moment at which the timing is different from the intended timing is at 500 ms, which therefore cannot have impacted the article-N400s. This is what we meant by “after the N400 window”: this is a calculation, not an assumption. We have removed the references about slower reading times leading to more predictive processing to avoid further confusion.

5) The use of different baselines was confusing to multiple reviewers. A table that summarizes details of all the current analyses and how these correspond to the procedures used in the original DUK study would help the reader keep straight the different results.

There are only three different baseline procedures: none, 100ms or 500 ms. We used different baselines because DUK did not correctly report their procedure. We have explained and motivated our pre-registered baseline procedures and motivated changes to the baseline procedure in exploratory analyses. We have also changed “Original baseline procedure” in Figure 6 to avoid confusion, because this is ambiguous (it could mean what was reported or what was done).

All the procedures are summarized in the table 1. We do not think it is necessary to include it in the main text but will include it if the editor deems it important enough to include.

Table 1. Reported or used baseline correction procedures in DeLong, Urbach and Kutas (2005) and in the current studyOriginal studyCurrent studyCorrelation analysisReported no baseline correctionPre-registered no baseline correction (Figures 1 and 5)Later acknowledged using a 500 ms baseline correctionRe-analysis with a 500 ms baseline correction (Figures 2 and 6)Single-trial analysisPre-registered a 100 ms baseline correction (Figures 3 and 4)Re-analysis with a 500 ms baseline correction (Figures 5 and 7)

6) Both the original and current studies present a high percentage of a/an continuations in the cloze test. Could this procedure bias participants towards using a/an rather than "his", "hers", "their," etc. that aren't marked for the phonological form of the subsequent noun? After encountering several sentences in which the indefinite article is appropriate, participants may routinely complete the remaining sentences with an indefinite article continuation due to this sort of syntactic priming. A comment about this would be welcome.

Inspection of the raw cloze results suggests that this is a very unlikely scenario. We suspect that the original items were designed to elicit a/an responses instead of possessive pronouns, which would make sense given the goal of the experiment, but we cannot be certain, and, to our knowledge, the raw cloze data of DUK are not publicly available. In our own cloze data, there are two items where several participants use possessive pronouns (for the famous kite/airplane example item number 8, and another item with ‘her finger’ which is number 50 in the list. So, our participants did use possessive pronouns when those pronouns came to mind as best completions even more than halfway through the list.

7) In multiple places, the authors use phrasing about how the study argues for "a more limited role for prediction during language comprehension." This wording seems overly broad, conflating very different conceptions of prediction across different levels of representation. There is a deep theoretical division between proposed mechanisms of conceptual pre-activation (e.g., the concept of 'readiness' in the discourse literature, linked to the N400 literature beautifully in Jos Van Berkum's work) and mechanisms of predicting the form of the upcoming input (commitment to a particular linguistic formulation of the message). It seems likely that the former, long argued for in the discourse literature, might play a central role in language comprehension, while contributions from the prediction-of-form mechanism may be quite limited. It would be unfortunate if readers take the current study to cast doubt on both extremely different mechanisms, when it actually only speaks to the latter mechanism. Please consider rewording this throughout.

We have removed or reworded that phrase throughout the text.